# 3D printable and biocompatible PEDOT:PSS-ionic liquid colloids with high conductivity for rapid on-demand fabrication of 3D bioelectronics

Byungkook Oh[1,9], Seunghyeok Baek[1,9], Kum Seok Nam [2,9], Changhoon Sung [2], Congqi Yang[2], Young-Soo Lim[3], Min Sang Ju[4], Soomin Kim[1], Taek-Soo Kim [4], Sung-Min Park [3,5,6,7], Seongjun Park [1,2,8] ✉ & Steve Park [1,8] ✉

3D printing has been widely used for on-demand prototyping of complex three-dimensional structures. In biomedical applications, PEDOT:PSS has emerged as a promising material in versatile bioelectronics due to its tissue-like mechanical properties and suitable electrical properties. However, previously developed PEDOT:PSS inks have not been able to fully utilize the advantages of commercial 3D printing due to its long post treatment times, difficulty in high aspect ratio printing, and low conductivity. We propose a one-shot strategy for the fabrication of PEDOT:PSS ink that is able to simultaneously achieve on-demand biocompatibility (no post treatment), structural integrity during 3D printing for tall three-dimensional structures, and high conductivity for rapid-prototyping. By using ionic liquid-facilitated PEDOT:PSS colloidal stacking induced by a centrifugal protocol, a viscoplastic PEDOT:PSS-ionic liquid colloidal (PILC) ink was developed. PILC inks exhibit high-aspect ratio vertical stacking, omnidirectional printability for generating suspended architectures, high conductivity (~286 S/cm), and high-resolution printing (~50 μm). We demonstrate the on-demand and versatile applicability of PILC inks through the fabrication of 3D circuit boards, on-skin physiological signal monitoring e-tattoos, and implantable bioelectronics (opto-electro-corticography recording, low voltage sciatic nerve stimulation and recording from deeper brain layers via 3D vertical spike arrays).

Three-dimensional (3D) printing has been widely adopted for commercial use due to its customizability, fast production cycle, and ability to create complex three-dimensional geometries[1–5]. Such properties of 3D printing have led to its wide usage in biomedical applications for the multi-material patterning of tissue-like materials[6,7], such as conducting polymers and hydrogels, into complex anatomical structures[8] and bioelectronic devices[9]. As a result, these soft bioelectronic interfaces have minimized tissue damage and enabled the fabrication of patient specific devices for increased therapeutic efficacy[6,7].

For bioelectronics devices, conductive polymers such as poly(3,4-ethylenedioxythiophene):polystyrene sulfonate (PEDOT:PSS) have emerged as promising materials due to its electrical interfacing properties with tissues, tissue-like mechanical properties, and biocompatibility[10,11]. However, pristine PEDOT:PSS has low yield stress and storage modulus, making it incompatible with 3D printing. In

order to improve the printability of PEDOT:PSS, additives such as dimethylsulfoxide (DMSO) and ionic liquids have been utilized[10,12]. However, the addition of such cytotoxic materials requires lengthy removal procedures after printing, such as repeated washing or annealing cycles, preventing immediate usage[10,12–17]. Ideally, the printed bioelectronic ink should be highly conductive, possess sufficient structural integrity for the versatile fabrication of various 3D architectures (e.g., vertically stacked or suspended structures), and be immediately usable without the need for complex post-processing. Such features have yet to be realized for conductive polymers.

In this work, we present a one-shot strategy for the fabrication of a biocompatible, highly conductive, and 3D printable PEDOT:PSS-ionic liquid colloidal (PILC) ink that allows on-demand fabrication of bioelectronics as depicted in Supplementary Fig. 1 and described in the

Methods "section" in detail. Through the use of a centrifuge-based strategy, the ionic liquid acts as a catalyst to facilitate an ionic exchange reaction. This results in the formation of a hydrogen-bonded network of densely packed PEDOT colloids, referred to as the PILC ink (Fig. 1a). The PILC ink exhibits high conductivity (286 S/cm)[10,12,16,18–20] (Supplementary Fig. 2) in comparison to pristine PEDOT:PSS (~1 S/cm)[6], making it suitable as an electrode material. Furthermore, excess ionic liquid and PSS are effectively removed from the PILC ink during centrifugation, resulting in a biocompatible ink that does not require any post-treatment processes. Moreover, the dense colloidal packing in the PILC ink results in high yield stress and self-supporting mechanical rigidity during 3D printing (i.e., storage modulus of $10^5$ Pa, and yield stress of $10^3$ Pa) (Fig. 1b). This results in high structural integrity upon PILC ink extrusion, in contrast to conventional

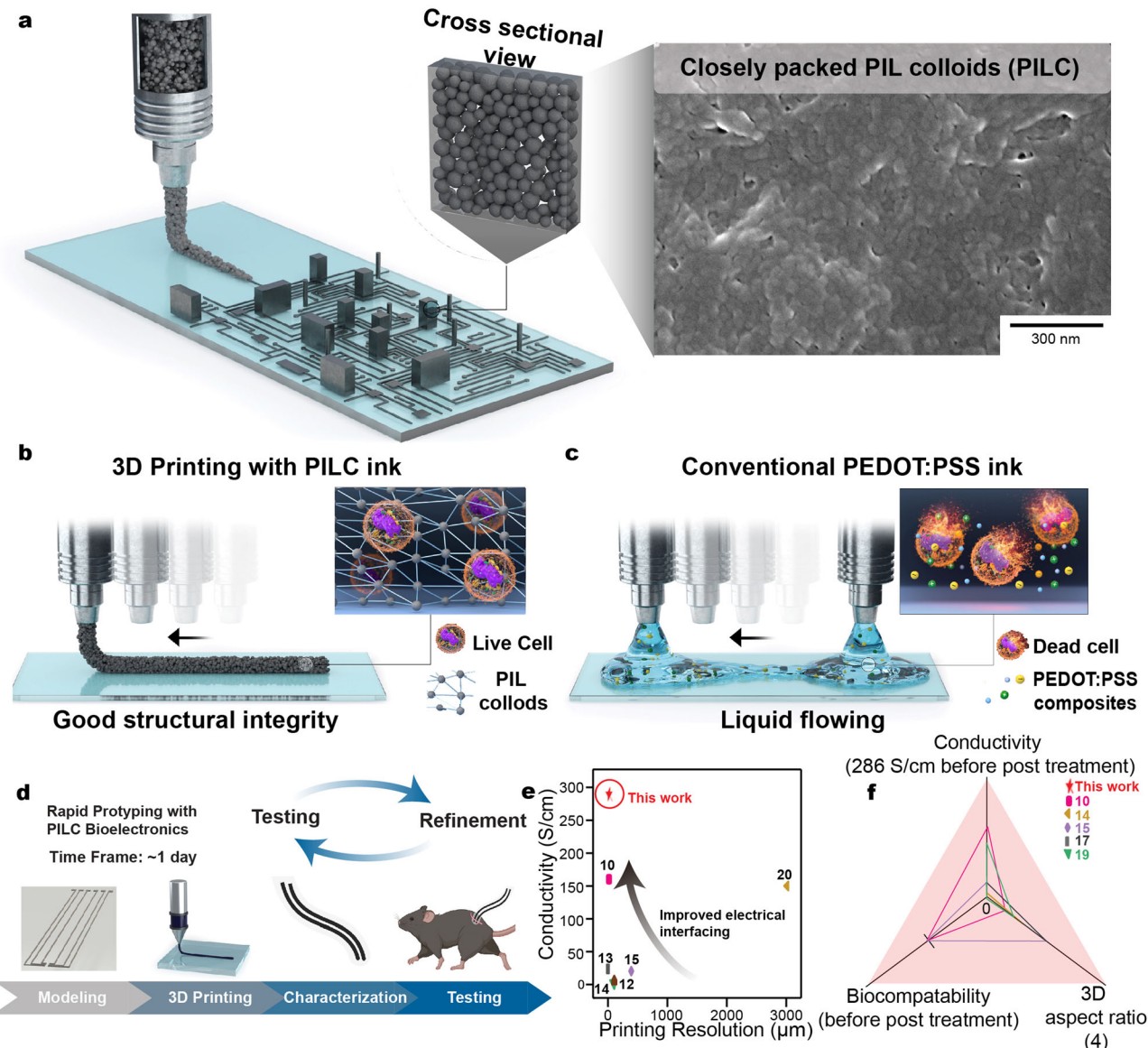

**Fig. 1 | PEDOT:PSS ionic liquid colloidal (PILC) ink for 3D printed bioelectronics. a** Schematic illustration (left) and SEM image (right) of the PILC ink enabling 3D printed structures with high structural integrity through the dense packing of colloidal particles. Data reproducibility was confirmed by three independent experiments. **b** Schematic illustrations of the PILC ink with minimal ionic liquid content and good structural integrity. **c** Schematic illustrations of conventional PEDOT:PSS ionic liquid composites with cytotoxic ionic liquid components and poor structural integrity. **d** Schematic illustration of the PILC ink for rapid prototyping of PILC bioelectronics. **e** Conductivity, and printing resolution of the PILC ink in comparison to previous works using 3D printable PEDOT:PSS inks. **f** 3D aspect ratio, conductivity and biocompatibility of the PILC ink before post treatment in comparison to previous works using 3D printable PEDOT:PSS inks. Figure 1/ panel d Created with BioRender.com released under a Creative Commons Attribution-NonCommercial-NoDerivs 4.0 International license.

PEDOT:PSS inks that are limited in 3D stacking due to its rheological properties (Fig. 1c). This enables the fabrication of highly stacked 3D structures, high resolution structures (50 μm), and suspended structures (2 mm overhang). In addition, the high conductivity and immediate usability of the printed PILC ink enable rapid usage in diverse biomedical applications, ranging from efficient physiological signal recording (EMG and ECG) via on-skin e-tattoos, to implantable bioelectronics for opto-electrocorticography (ECoG) recording and low-voltage stimulation (60 mV) of the sciatic nerve in vivo (Fig. 1d). The combination of unique material properties (high conductivity, high printing resolution, biocompatibility before post processing, and high 3D aspect ratio) exhibits the broad applicability of the PILC ink as a general purpose PEDOT:PSS ink in bioelectronic applications (Fig. 1e, f, Supplementary Fig. 3 and Supplementary Table. 1–3).

## Results

### PILC Ink Design Principles

PEDOT:PSS has been widely reported to exist in a colloidal state in aqueous solution. While previous literature has utilized additives to induce interconnected PEDOT chain networks and thereby improve rheological properties for 3D printing[10], their rheological properties (i.e., low yield stress and storage modulus) have limited the printing of high aspect-ratio structures. In contrast, recently reported materials have utilized densely packed microdroplets to increase storage modulus and enable omnidirectional printing[21]. In order to create a PEDOT:PSS ink with structural stability for tall 3D structures, we hypothesized that the dense packing of PEDOT:PSS colloidal particles on nanometer-length scales can induce similar rheological properties.

To achieve the above, we utilized ionic liquids to obtain the dense packing of PEDOT:PSS colloids in aqueous solution (deionized water). The ionic liquid was hypothesized to play a three-part role: 1) improve the conductivity[22], 2) induce ink formation through phase separation, and 3) enhance hydrogen bonding between colloidal particles through an ionic exchange reaction for improved rheological properties (Fig. 2a)[23–25]. We utilized EMIM:TCB (1-Ethyl-3-methylimidazolium tetracyanoborate) as the ionic liquid due to its ability to create significant increases in conductivity[24]. By adding EMIM:TCB during ink fabrication, PEDOT:PSS demonstrated phase separation after centrifugation (Supplementary Fig. 4). In contrast, when no ionic liquid is used or DMSO is utilized, no phase separation is observed, demonstrating the key role of the ionic liquid in ink formation. Furthermore, adding other ionic liquids (1-Ethyl-3-methylimidazolium ethyl sulfate (EMIM:ES) and 1-hexyl-3-methylimidazolium tetracyanoborate (HMIM:TCB)) during ink fabrication did not induce phase separation after centrifugation (Supplementary Fig. 5). To test the hypothesis on ionic liquid facilitated-hydrogen bonding, FT-IR spectra were measured as a function of PILC ink drying time (thus at different amounts of solvent in the film) (Fig. 2b). As drying time increases, the −OH vibration peak progressively shifted to a lower wavenumber, indicating that the hydrogen bonded network is increasing[26,27].

### PILC Ink Characterization

In order to investigate the compatibility of the PILC ink for 3D printing, we evaluated the rheological properties and 3D patternability. The dense colloidal microstructure of the PILC ink improved the structural stability of the extruded ink as observed through its increased viscosity (~1 MPa·s at 0.1 Hz) and yield stress (1 kPa at 0.01 Hz) in comparison to the viscosity (~0.1 kPa·s at 0.1 Hz) and yield stress (47.4 Pa at 0.01 Hz) of pristine PEDOT:PSS (Fig. 2c, d). The pristine PEDOT:PSS ink has low viscosity at all shear rates, which renders it non-printable. In contrast, PILC inks shows shear thinning behavior with high viscosity at low shear rates by enhancing hydrogen bonding between PEDOT:PSS-ionic liquid colloidal particles, along with long term ink stability (Supplementary Fig. 6). In addition, the pristine ink shows Newtonian behavior; whereas the PILC ink exhibits rheopectic behavior, with high

shear stresses at low shear rates, demonstrating why the PILC ink can maintain its structure after printing[28]. To analyze the source of enhanced shear thinning in PILC inks, urea was used to disrupt hydrogen bonding[29]. The shear thinning effects of PILC inks decreased with increasing urea molar ratio as urea acts to disrupt hydrogen bonding between interconnected PEDOT chains (Supplementary Fig. 7).

The PILC ink rheological properties can be tuned for various printing conditions by varying the PILC pellet concentration (Supplementary Fig. 8). Low concentration PILC inks are optimal for high resolution printing; whereas, high concentration PILC inks have increased yield stress for the fabrication of highly stacked structures. This enables the rapid prototyping of different device designs based on the required specifications. The PILC ink can be used to print high resolution structures (~50 μm line width) (Fig. 2e, and Supplementary Fig. 9 and Supplementary Video 1) due to its small average particle diameter of 18.8 nm and standard deviation of 8.1 nm as measured through cryo-transmission electron microscopy (cryo-TEM).

From its viscoplastic behavior, the PILC ink demonstrated universal 3D printing. As the PILC ink acts as a solid after extrusion at low shear rates, it is less limited by liquid-state printing behavior such as wettability. The PILC can be printed on a variety of substrates regardless of surface hydrophilicity or hydrophobicity such as hydrogels, silicone, porcine skin (Supplementary Fig. 10). In addition, PILC ink can be utilized to print tall, customized three-dimensional structures. The PILC can be highly stacked with an aspect ratio of 4, which is relatively high compared to previous PEDOT:PSS-based inks, and can be fully dried within 1 min at 60 °C (Fig. 2f–h).

The PILC electrode also demonstrated high conductivity (286 S/cm) through the addition of the EMIM:TCB. The interaction between EMIM:TCB and PEDOT:PSS has been shown to increase conductivity by separating PEDOT and PSS and enabling the effective removal of the insulating PSS layer[24]. This mechanism was confirmed in the PILC ink through X-ray photoelectron spectroscopy (XPS) and Raman spectroscopy as seen in Supplementary Fig. 11. The PEDOT to PSS ratio can be calculated by integrating the PEDOT (162 - 167 eV) and PSS peak (166 - 171 eV) of the S $2p$ XPS spectra[30,31]. The PILC electrode has a high ratio of the PEDOT phase (PEDOT/PSS = 1.57) compared to pristine PEDOT:PSS (PEDOT/PSS = 0.46) due to the extraction of PSS in the centrifuge process, explaining the increased conductivity of the PILC electrode. Furthermore, the Raman spectrum indicates that the PEDOT chains in the PILC electrode undergo a structural transition from a benzoid form to quionoid form[32]. This is induced by the ionic exchange reaction between the ionic liquid and PEDOT:PSS and contributes to the high conductivity in the PILC electrode[24].

The PILC electrode also demonstrates immediate biocompatibility without any post-treatment through the centrifugal phase separation of the cytotoxic ionic liquid (Fig. 3a). Firstly, the remaining presence of cytotoxic EMIM:TCB was measured in the pellet. SEM-EDS mapping of the PILC electrode demonstrated a markedly reduced presence of boron and nitrogen atoms in comparison to pristine PEDOT:PSS ionic liquid composites (Fig. 3b, c). Furthermore, XPS analysis revealed the presence of EMIM:TCB in the supernatant solution (Supplementary Fig. 12), demonstrating the centrifugal separation of the cytotoxic components.

Subsequently, biocompatibility was evaluated with an in vitro MTT assay with NIH/3T3 fibroblast cells. The PILC electrode demonstrated decreased cytotoxicity compared to pristine PEDOT:PSS ionic liquid composites and cytotoxicity similar to the control group (Fig. 3d). An in vitro LIVE/DEAD assay further demonstrated high cell viability of PILC electrodes (Fig. 3e). To further validate the role of centrifugal phase separation on biocompatibility, cell viability was measured as centrifuging was repeated. Increasing cell viability was observed as centrifuging was repeated with a cell viability of 92 % after two centrifuging steps (Supplementary Fig. 13).

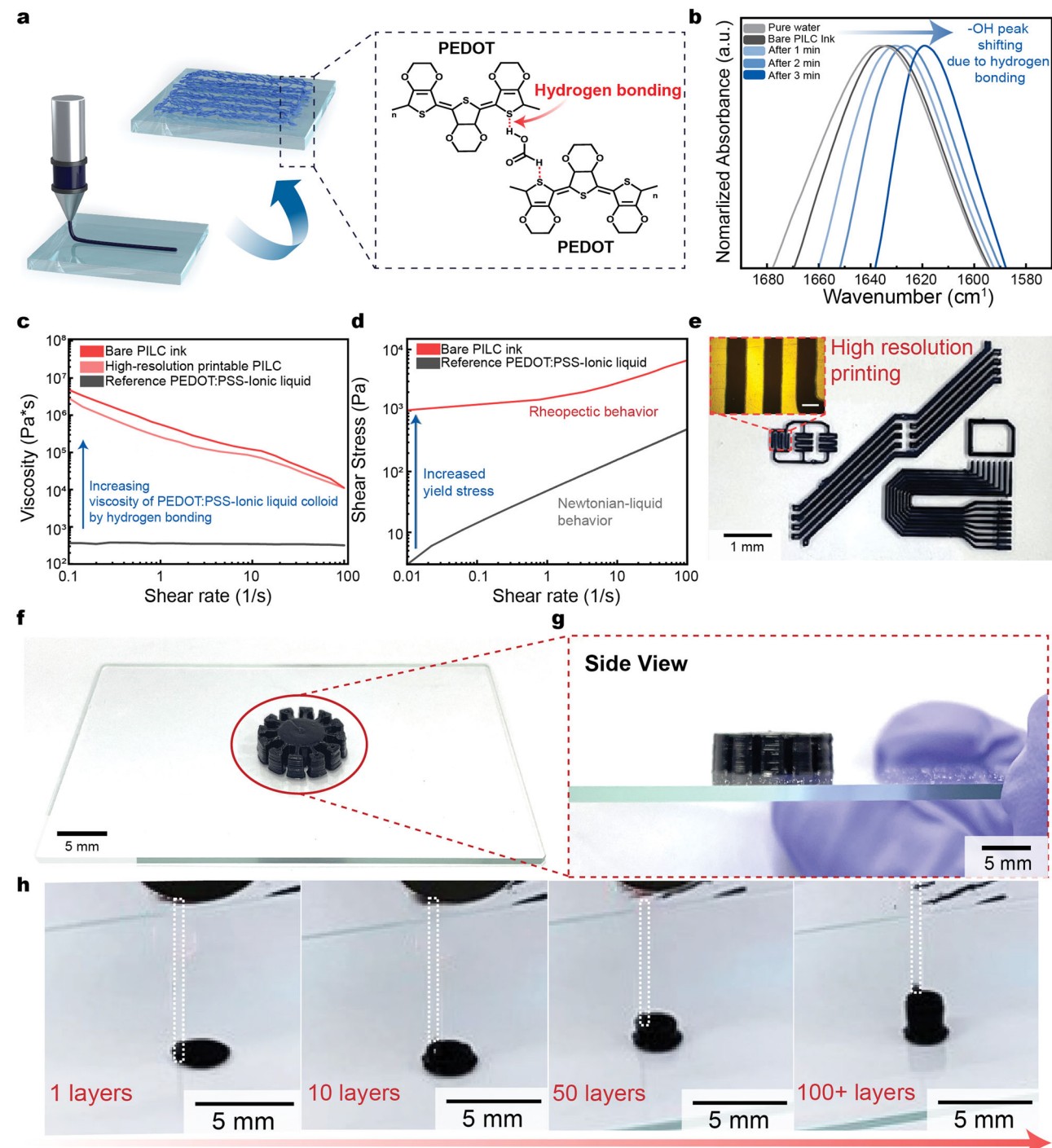

**Fig. 2 | Ionic liquid facilitated hydrogen-bonding of PILC ink for high resolution and high aspect ratio 3D printing. a** Schematic illustration of the PILC ink formed through ionic liquid facilitated hydrogen bonding. **b** FT-IR spectra of the PILC ink as hydrogen bonding progresses (measured while drying the PILC ink). **c** Viscosity of PILC ink in comparison to conventional PEDOT:PSS ionic liquid composites. **d** Shear stress of PILC ink in comparison to conventional PEDOT:PSS ionic liquid composites. Increased yield stress is shown to indicate the appearance of rheopectic behavior. **e** Image of high-resolution printing (~50 μm) of the PILC ink. Scale bar: 50 μm. **f**–**g** Images of a multilayer Ferris's wheel structure from top-front (**f**) and side (**g**) views. **h** Images of the PILC ink during the printing of multilayer structures.

## On-demand Applications of the PILC Ink

The aforementioned rheological properties of the PILC ink enable omnidirectional printing of three-dimensional circuit boards (Fig. 4a). The PILC ink can be utilized to print suspended interconnects by simultaneously moving the x- and z- stage in the air (Fig. 4b, c). In addition, 3D diagonally suspended circuit lines can be printed onto an artificial kidney (made of Ecoflex-0020, with a height of 7 mm), demonstrating the potential of employing a 3D interconnection of PILC inks via omnidirectional printing. This approach can effectively bridge the two different interfaces, even with a high angle and gap, (as shown in Supplementary Fig. 14). The PILC ink demonstrates increased yield stress and storage modulus in comparison to previously reported materials[10,12,17,19], which was key for enabling the omnidirectional printing (Fig. 4d−f)[21]. Furthermore, the yield stress and storage modulus can adjusted by changing the dilution ratio of the PILC ink (Supplementary Fig. 15). Dilution ratios of <60% are required for high

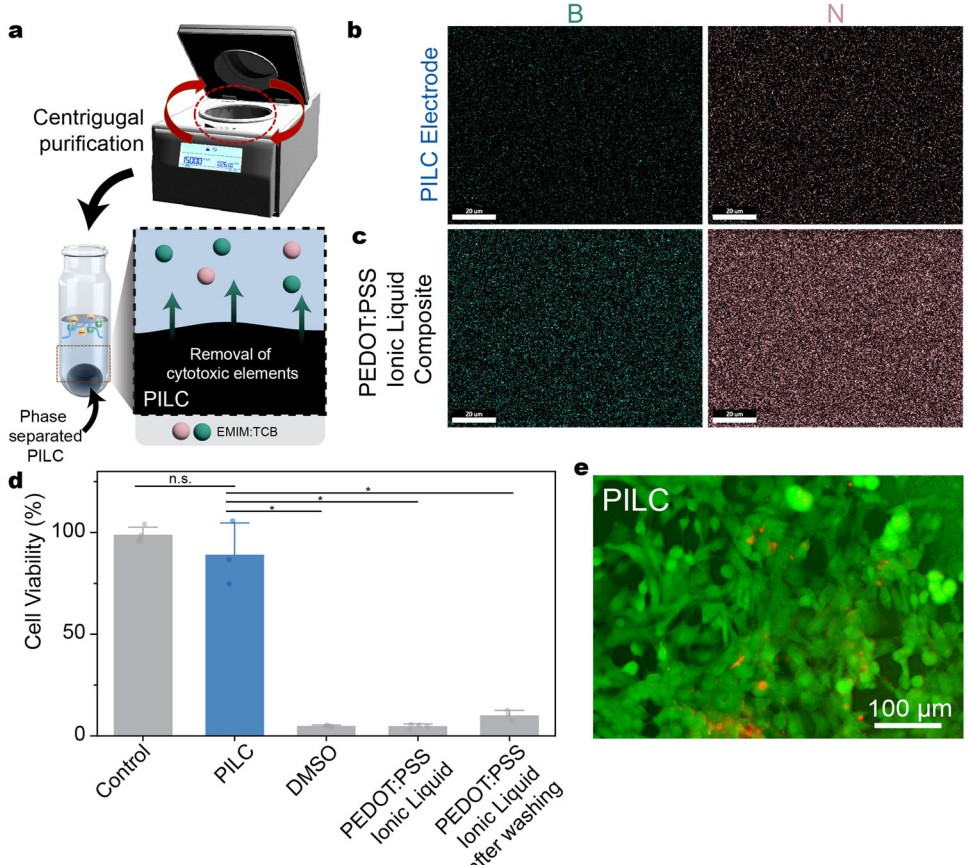

**Fig. 3 | Biocompatibility of the PILC ink through centrifugal removal of ionic liquids. a** Schematic illustration of centrifugal removal of ionic liquid in the PILC ink. **b**, **c** SEM-EDS mapping of key elements of EMIM:TCB (boron and nitrogen) in (**b**) PILC electrodes and (**c**) PEDOT:PSS ionic liquid composites. Data reproducibility was confirmed by three independent experiments. **d** Cell viability of PILC electrodes in comparison with PEDOT:PSS ionic liquid composites before and after washing. PILC vs Control: P = 0.39175, t = −1.06193, d.f. =2.17694; PILC vs DMSO: P = 0.01102, t = 9.33251, d.f. =2.01484; PILC vs PEDOT:PSS Ionic Liquid: P = 0.01124, t = 9.33642, d.f. = 2.00211; PILC vs PEDOT:PSS Ionic Liquid after washing:

P = 0.01124, t = −8.65617, d.f. =2.10067. Values in (**d**) represent the mean and standard deviation (*n* = 3 independent material samples for PILC and PEDOT:PSS Ionic Liquid after washing groups, *n* = 4 independent material samples for DMSO, PEDOT:PSS Ionic Liquid, and PEDOT:PSS Ionic Liquid after washing groups). Statistical significance was determined through a two-sided t-test between the PILC electrode and all comparison groups after Welch Correction; * *p* < 0.05, n.s. not significant. **e** LIVE/DEAD assay images of the PILC electrodes (*n* = 3 independent PILC samples).

storage modulus and yield stress values that provide appropriate structural integrity to enable omni-directional printing. In addition, the high conductivity of the PILC ink enables usage as an LED interconnect (Fig. 4g–i, Supplementary Video 2).

The improved yield stress and storage modulus of the PILC ink, compared to pristine PEDOT:PSS inks and other PEDOT:PSS composites, enable the 3D high aspect ratio printing of PILC inks without any collapse of printed structures during printing (Fig. 5a). Additionally, due to its high yield stress and storage modulus, the PILC ink can be printed in air without any support layers, enabling the creation of bridges with the potential for 3D interconnections of individual circuits (Fig. 5b). Leveraging these capabilities for 3D high aspect ratio and air printability, an LED chip was mounted on a pyramid structure created using PILC inks after printing (Fig. 5c) and subsequently operated (Fig. 5d).

The unique properties of the PILC ink also make it an ideal material for rapid on-demand fabrication of on-skin bioelectronics for health monitoring applications. As electrodes printed from PILC ink have similar mechanical properties (Young's modulus 750 kPa, Tensile Strength 139.3 kPa, and Toughness 4.0 kJ/m$^3$ as seen in Supplementary Fig. 16) with biological tissues[33], PILC electrodes can be utilized as soft electrodes to interface with tissue. Furthermore, printed PILC electrodes on soft substrates (PDMS) have stretchability at up to 90% strain

and cyclability at 70 % strain for 1000 cycles (Supplementary Fig. 17 and Fig. 18) in contrast to freestanding PILC film, which can only be stretched up to 10% strain. This can be attributed to the energy dissipation mechanism as previously reported[34,35]. We demonstrate the use of PILC ink to fabricate a large area of on-skin e-tattoo (Supplementary Fig. 19a). After printing the PILC ink on a super hydrophobic (polypropylene) flexible substrate, the PILC electrodes can be transferred onto wet skin under gentle pressure (Supplementary Fig. 19a, Supplementary Fig. 20) with an adhesion strength of 1.884 N/mm between wet porcine skins and printed PILC films, which is needed to peel off wet porcine skins attached with a film of printed PILC (Supplementary Fig. 21). The high conductivity and conformal contact of PILC electrodes results in lower interfacial skin impedance (200 kΩ mm$^2$ at 1 kHz) compared to that of the commercial 3 M electrodes (8000 kΩ mm$^2$ at 1 kHz) (Supplementary Fig. 19b).

Utilizing the lower interfacial skin impedance, PILC electrodes demonstrate high-quality recordings of both ECG (electrocardiogram) and EMG (electromyogram) waveforms. PILC electrodes demonstrated an increased signal amplitude in the ECG waveform in comparison with commercial 3 M electrodes (Supplementary Fig. 19c). In addition, PILC electrodes demonstrate high signal-to-noise ratios (16.80 dB) during EMG recordings of bicep muscles (Supplementary Fig. 19d).

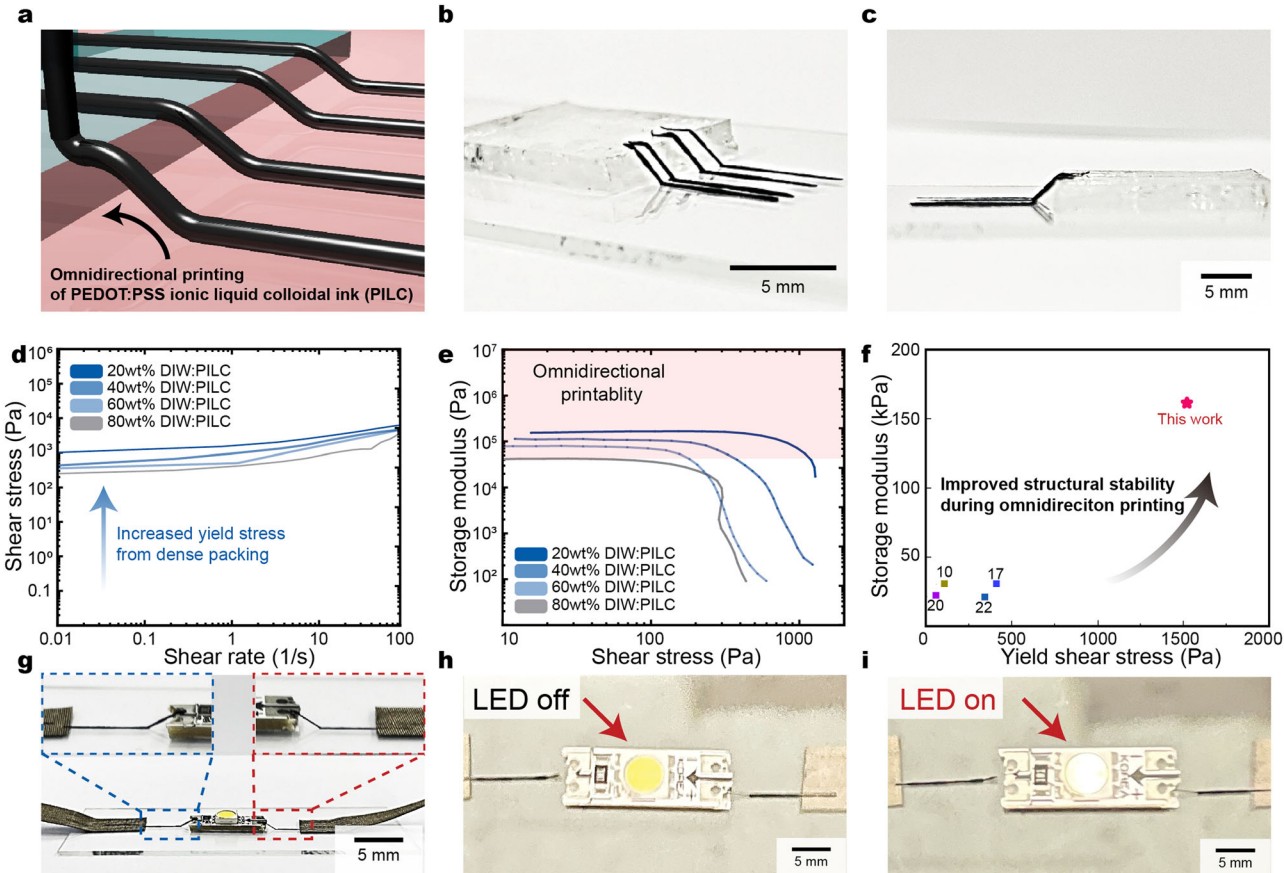

**Fig. 4 | PILC ink enabled omnidirectional printing of 3D circuits. a** Schematic illustration of the PILC ink for the omnidirectional printing of conductive interconnects. **b, c** Top-side view (**b**) and side view (**c**) images of printed PILC inks with large overhangs for 3D circuits (**d**) Shear stress of optimized PILC ink for omnidirectional printing. **e** Storage modulus of optimized PILC inks for omnidirectional printing. Red shaded area marks necessary storage modulus for stable omnidirectional printing. **f** Storage modulus of the PILC ink in comparison with previously reported PEDOT:PSS inks for 3D printing. **g–i** Images of the printed PILC ink for 3D connection to an LED (**g**). Images of the LED turned off (**h**) and on (**i**).

In addition to on-skin bioelectronics, the PILC ink can be utilized to fabricate PILC devices for implantable bioelectronics. In order for PILC devices to be utilized in vivo environments, the PILC electrodes should be able to handle mechanical and electrical stress during surgical insertion and potential animal motor responses (i.e., leg movement). To verify the mechanical stability, a cyclic bending test is conducted with a bending radius of 1.5 mm (Fig. 6a). The PILC electrode demonstrates minimal change in resistance during 10,000 bending cycles, demonstrating its tolerance to mechanical deformation during in vivo electrode applications. Furthermore, the PILC electrical properties, demonstrate high conductivity in saline and tolerance to wet physiological conditions, enabling its application for electrophysiological recording (Fig. 6b and Supplementary Fig. 22). In addition, the high charge storage capacity of the PILC electrode in comparison to conventional metals, such as gold, demonstrates its efficient charge injection in physiological environment and opens potential for utilizing lower stimulation voltages, and thereby safer stimulation conditions in vivo (Fig. 6c and Supplementary Fig. 23). To evaluate the feasibility of PILC devices for implantable bioelectronics, we performed distinct applications in an in vivo mouse model (Fig. 6d and Supplementary Fig. 24). The high conductivity and high-resolution printing (line width and pitch of 50 and 100 μm, respectively) of PILC inks enabled its use in optogenetic ECoG devices, while its high CSC also made it suitable for sciatic nerve stimulation devices. A multi-channel PILC ECoG device enabled the recording of optically-evoked ECoG signals in a Thy1-Chr2 transgenic mouse with high SNR (SNR 9.0) (Fig. 6e, f). Furthermore, the PILC ECoG device demonstrates high

spatial-selectivity with a high signal amplitude at the site of optical stimulation (-290 μV at an electrode 0 mm from optical stimulation) compared an adjacent electrode (−56 μV at an electrode 1.1 mm from optical stimulation) (Supplementary Fig. 25). Moreover, the printed PILC devices enabled low-voltage stimulation (-60 mV) of the sciatic nerve, demonstrating its use for safe electrical stimulation in vivo (Fig. 6g, h and Supplementary Fig. 26). The unique properties of the PILC ink (biocompatibility without postprocessing, high structural integrity, and high conductivity) enable its use as a general purpose PEDOT:PSS ink.

We also demonstrated that the versatile printing of the PILC ink enables the fabrication of 3D bioelectronic devices, such as vertical spike arrays (Fig. 7a, b). These flexible and biocompatible devices, consisting of an array of PILC electrodes with high aspect ratio, enable the recording of neural signals from deep brain regions, such as the hippocampus, which is difficult to achieve with 2D surface arrays. We validate that the recorded signals demonstrate characteristics of hippocampal signals by examining the signal frequency spectrum. A dominant signal characteristic of the hippocampus is theta wave oscillations, which are signals prominent in the 4 - 8 Hz frequency band[36]. The 3D PILC arrays demonstrates peaks at ~5 Hz and its harmonic frequencies as shown in the power spectral density (Fig. 7c), spectrogram (Fig. 7d) and raw signal waveforms (Fig. 7e). While in contrast, this peak is not observed in the 2D PILC arrays, due to their spatial restriction to the surface of the brain (Fig. 7c–e).

In addition, the 3D PILC arrays demonstrate high fidelity recording during the electrophysiology of sensory evoked potentials (Fig. 7f).

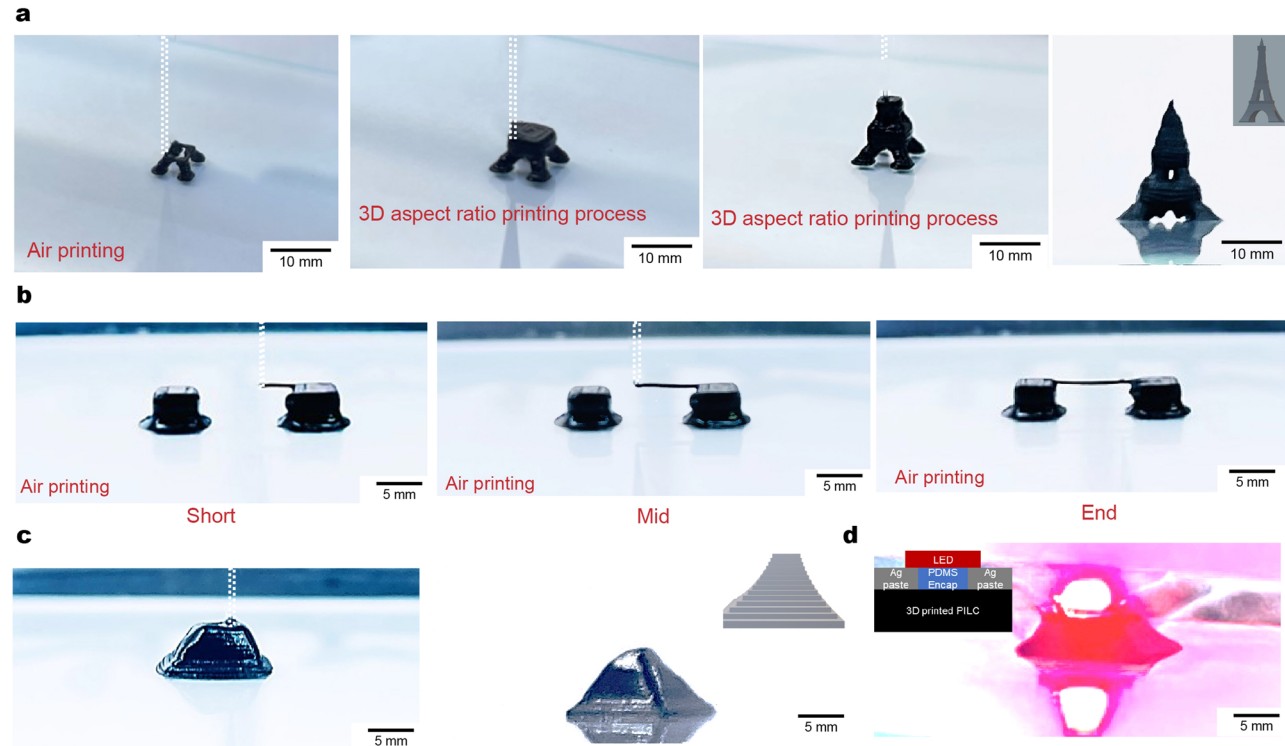

**Fig. 5 | 3D high aspect ratio printing of PILC inks with air printing w/o supported layers. a** Images of 3D high-aspect ratio printing of PILC inks. **b** Images of air printing of PILC inks without supported layers. **c** Images of 3D printed PILC structure that mimics pyramid structures. **d** Images of an LED mounted on the 3D printed PILC structure.

Cortical neural signals demonstrate maximal amplitude when neural probes are inserted 300 ~ 500 μm below the cortical surface (i.e., Layer IV ~ V)[37]. We target the secondary visual cortex, which has been shown to have reduced signal amplitude in comparison to the primary visual cortex upon application of visual stimuli.

Upon the application of visual stimulus to the eye contralateral to the PILC array, signals-correlated to stimulus intensity were observed in the 3D PILC array in contrast to the 2D array (Fig. 7g, h). The correlation in visually-evoked signal amplitude and stimulus intensity verifies successful recording of sensory evoked potentials. In addition, to verify that the signal is not a recording artifact, ipsilateral visual stimulation is also conducted. No signal is observed upon ipsilateral stimulation, suggesting there is minimal light-induced artifacts (Fig. 7i).

## Discussion

We developed a one-shot strategy to fabricate 3D printable, biocompatible, and highly conductive PEDOT:PSS-ionic liquid colloidal (PILC) inks that can be used in various bioelectronics applications. By using ionic liquid-facilitated phase separation, PILC inks demonstrate high conductivity without post processing, and favorable rheological properties for the printing of tall three-dimensional structures. In addition, the centrifuging process removes excess ionic liquid to enable biocompatibility without post-processing, and chemical stability. The versatile printing of the PILC ink enables broad applicability ranging from three-dimensional circuit boards and on skin bioelectronics for healthcare monitoring, to implantable bioelectronics with high customizability. The unique properties of the PILC ink demonstrate its potential for use as a general purpose PEDOT:PSS ink compared to currently available 3D printing inks requiring lengthy post treatment which is preventing immediate usage. The rapid on-demand fabrication of tissue-like PILC bioelectronics acts as a platform to accelerate advancement in probing the body for future disease treatment.

## Methods

### Materials

PEDOT:PSS aqueous solution (1.0 ~ 1.3 wt%) was purchased from Heraeus Electronic Materials. EMIM:TCB (1-Ethyl-3-methylimidazolium tetracyanoborate), TWEEN 80, and PBS (phosphate-buffered solution) were obtained from Sigma-Aldrich. EMIM:TCB was refined before use, wherein it was mixed with water (1:20 vol%) and subsequently subjected to centrifuging at 3000 rpm (centrifuge speeds (relative centrifugal force, (RCF)): 1,006 g) for 20 minutes. The supernatant was collected and heated at 155 °C for 2 days to extract pure EMIM:TCB. The relative centrifugal force (RCF) was calculated as follows:

$$RCF\,(g) = 11.2 * rotor\,radius * \left( \frac{revolutions\,per\,minute\,of\,rotor(RPM)}{1000} \right)^2 \quad (1)$$

### Fabrication of freeze-dried PEDOT:PSS-ionic liquid composites

PEDOT:PSS solution, water, and refined EMIM:TCB were mixed in a ratio of 2:1:0.03 wt% and stirred at 900 rpm for over 12 hours. Subsequently, the PEDOT:PSS-ionic liquid composite was immersed in liquid nitrogen for 10 minutes and freeze-dried for 72 hours.

### Fabrication of water-insoluble pure PEDOT:PSS

PEDOT:PSS solution and DMSO were mixed in a ratio of 2:1 wt% and stirred at 900 rpm for over 12 hours. Subsequently, the PEDOT:PSS DMSO composite was immersed in liquid nitrogen for 10 minutes and freeze-dried for 72 hours.

### Fabrication of PILC inks

10 ml of water and 200 mg of freeze-dried PEDOT:PSS-ionic liquid composite were subjected to tip-sonication for 2 hours at 30% amplitude (VC 505, Sonics & Materials, 3 mm microtip). Prior to tip-sonication, the freeze-dried PEDOT:PSS-ionic liquid composite was cut into

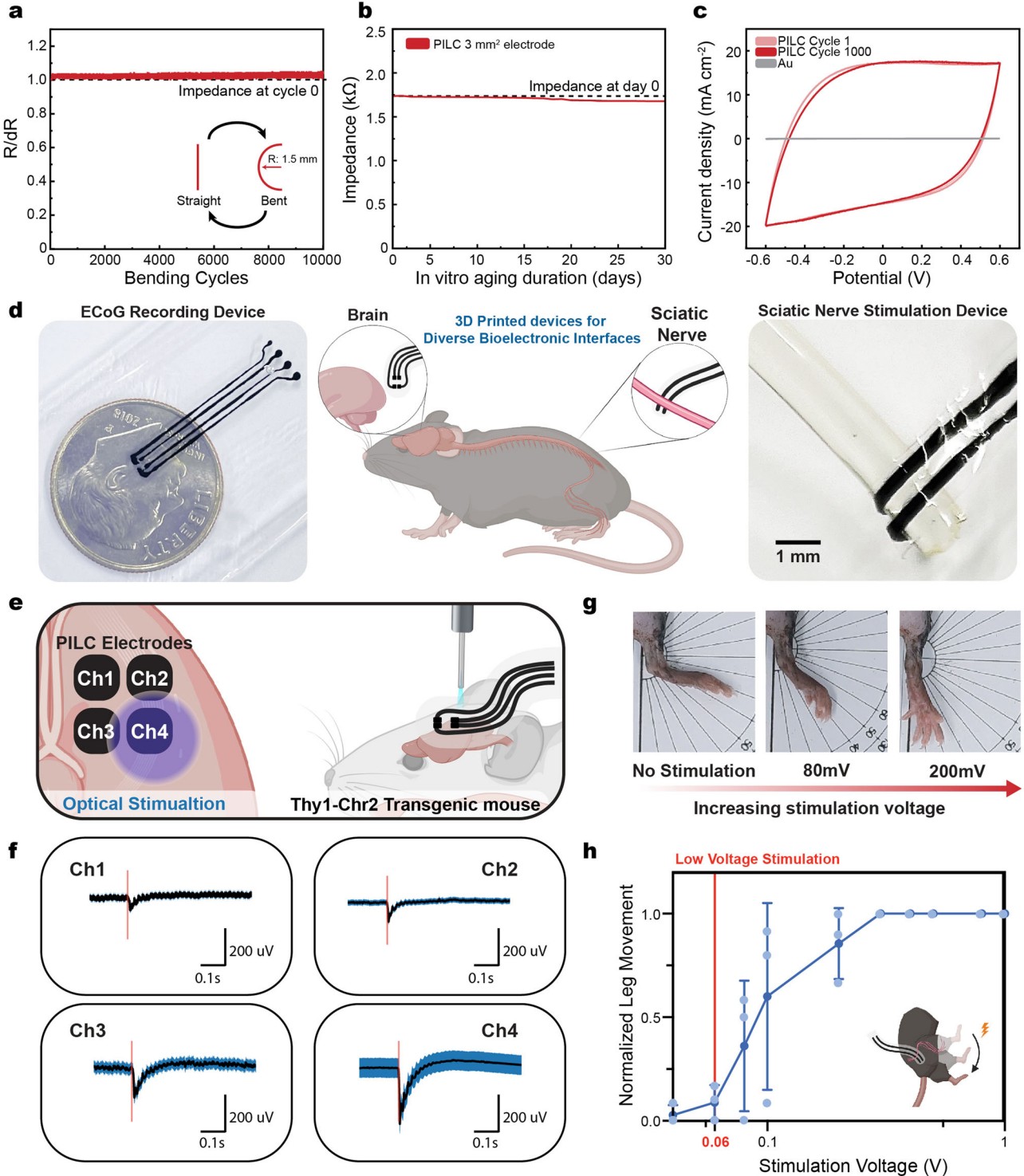

**Fig. 6 | 3D printed PILC devices for implantable bioelectronics. a** Change in resistance versus bending cycles of flexible PILC devices (Bending radius is 1.5 mm) **b** Impedance stability of PILC devices incubated in PBS solution at 37 C over 30 days. **c** Charge storage capacity of PILC devices across 1000 cyclic voltammetry cycles. Gold electrodes are plotted as reference. **d** Schematic (middle) and images of PILC implantable devices for optogenetic ECoG recording (left) and sciatic nerve electrical stimulation (right) **e** Schematic of optically-evoked ECoG recording in Thy1-Chr2 transgenic mice **f** Averaged waveforms of ECoG signals during optical stimulation in each channel of the multichannel device. Red vertical lines indicate the onset of optical stimulation pulses (5 ms pulse width). **g**, **h** Schematic illustration and images of the PILC device during electrical stimulation of the sciatic nerve. **g** Images demonstrate leg movement depending on stimulation voltage (**h**). Normalized leg movement as a function of stimulation voltage. 60 mV is marked to illustrate the onset of leg movement. Values in **g** represent the mean and standard deviation ($n = 3$ independent PILC samples). Figure 6/ panels **d**, **e**, and **h** Created with BioRender.com released under a Creative Commons Attribution-NonCommercial-NoDerivs 4.0 International license.

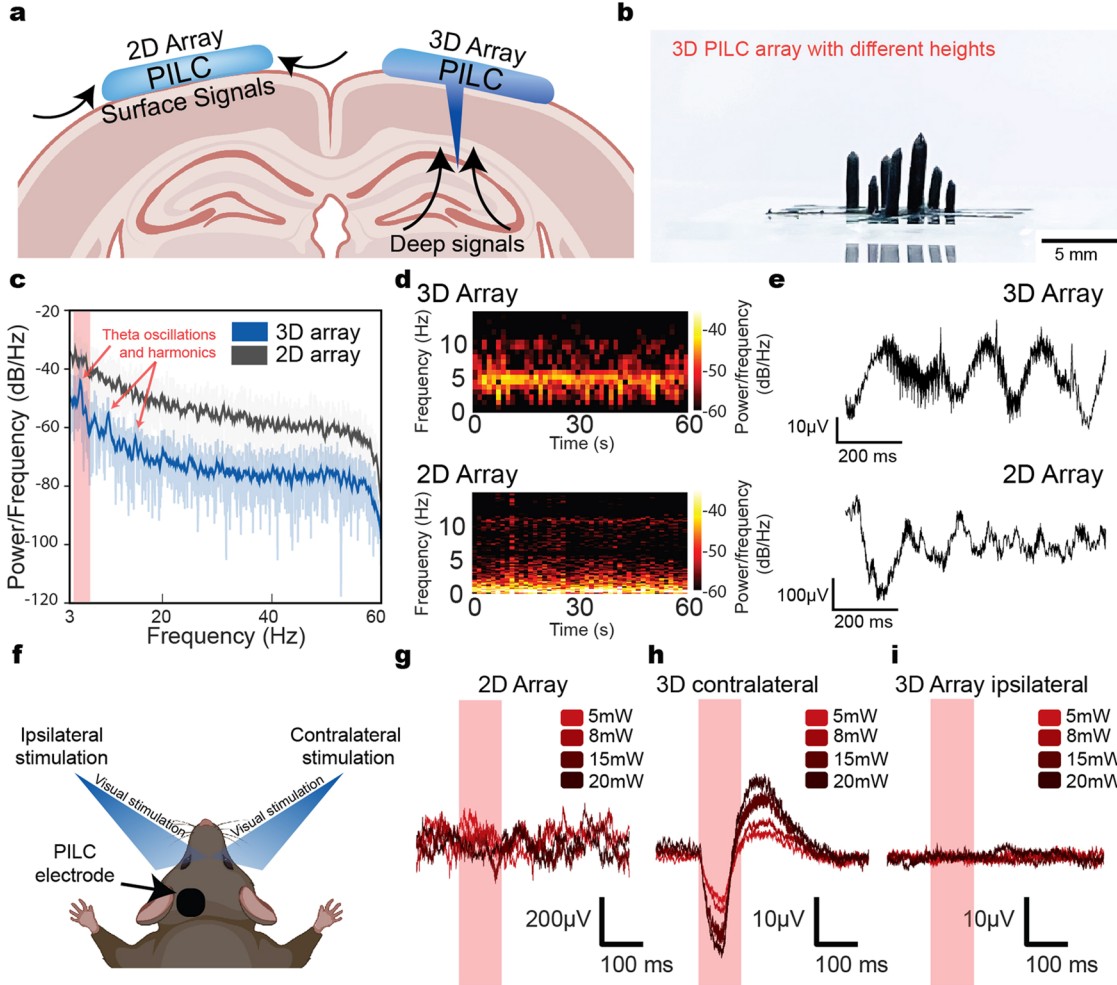

**Fig. 7 | 3D PILC devices for flexible in vivo bioelectronics. a** Schematic of 3D PILC arrays for fabrication of flexible, biocompatible 3D devices. 3D PILC arrays enable customizable interfacing with neural signals from deeper cortical layers and hippocampus, which are difficult to achieve with 2D surface arrays. **b** Image of 3D PILC arrays fabricated through the high aspect ratio printing of PILC ink **c** Power spectral density (PSD) of 3D and 2D PILC arrays for recording of characteristics endogenous neural signals from deeper brain regions (i.e., hippocampus). The red shaded area indicates the frequency band of theta oscillations (4 - 12 Hz). The theta band peaks and their harmonic frequencies in the PSD are marked with red arrows. **d** Spectrogram of 3D (top) and 2D(bottom) PILC arrays. **e** Raw local field potentials recorded from 3D PILC arrays (top) and 2D PILC arrays (bottom). **f** Schematic of sensory stimulation of the secondary visual cortex in an in vivo mouse model. **g–i** Evoked neural signals from a (**g**) 2D array during contralateral stimulation, (**h**) 3D array during contralateral stimulation, and (**i**) 3D array during ipsilateral stimulation. Figure 7/panels **a** and **f** Created with BioRender.com released under a Creative Commons Attribution-NonCommercial-NoDerivs 4.0 International license.

small pieces. Subsequently, 30 mg of TWEEN 80 was added, and tip-sonication was performed at 20% amplitude for 3 minutes. The resultant mixture was centrifuged for 45 minutes at 15,000 rpm (RCF: 25,155 g). After centrifugation, the supernatant is decanted, and 10 mL of water is added followed by thorough mixing with sediment. Subsequently, this solution was centrifuged under identical conditions, and the sediment pellet was utilized as the PILC 3D-printable ink. The rheological properties of PILC ink were adjusted through re-dispersion in water with a planetary mixer (Thinky AR-100) in mixing mode for 1 min.

### 2D & 3D printing of PILC inks
A direct-ink-writing printer (BIO X6, CELLINK) was used to print the PILC inks onto various films. The as-prepared PILC ink was loaded into a 3 mL syringe and utilized for the pressure-driven extrusion of PILC inks during printing. The tip diameter of the nozzles and printing speeds were determined in accordance with the specified requirements (nozzle diameter: 30 μm to 250 μm, printing speed: 6.4 mm/s to 12 mm/s). For the 3D printing of PILC inks, such as omnidirectional printing, MATLAB software (R2019a, The MathWorks, Inc.) was utilized

to customize the g-code and incorporated automatic x-stage movements (printing speed: 1.4 mm/s to 2.0 mm/s with 30 Gauge nozzles), z-stage movements (printing speed: 1.4 mm/s to 2.0 mm/s) and pressure control (50–100 kPa).

### Fabrication of 3D interconnects during omnidirectional printing
To demonstrate the fabrication of 3D interconnects, a SMD LED chip (5 V; GrinMax with a 1.5 mm overhang) was mounted on a glass substrate. The PILC ink was printed onto both the cathode and anode and connected to nickel conductive tape (3 M). To turn on the 3D interconnected LED, a voltage of 5 V was applied to the LED chip through a source meter (Keithley 2400, Tektronix Inc.).

### Fabrication of a 3D high aspect ratio circuit for operation of an LED chip
To demonstrate the fabrication of a 3D high aspect ratio circuit, a SMD LED chip (2.6 V; HSMH-H170 with a 7 mm of printed height) was mounted on a pyramid PILC structure. The PILC ink was connected onto both the cathode and anode with silver paste and connected to Teflon-wrapping wires (SME). To turn on the LED, a voltage of 2.6 V was

applied to the LED chip through a source meter (Keithley 2400, Tektronix Inc.).

## Fabrication of on-skin e-tattoo bioelectronics

The PILC ink was directly printed onto a super hydrophobic film (polypyrene) to fabricate on-skin e-tattoo bioelectronics. A custom design drawn via CAD software (Solidworks) was utilized. The printed PILC on-skin e-tattoo was utilized for monitoring physiological signals (EMG and ECG) after water-based transfer onto wet skin.

Raw data was filtered to eliminate noise and artifacts using Lab-Scribe. Both EMG and ECG signals underwent a 60 Hz notch filter to eliminate powerline noise. The EMG signal was further subjected to a 50 Hz high-pass filter and a 150 Hz low-pass filter to remove potential low-frequency motion artifacts while preserving the relevant EMG frequency range (50–150 Hz). The ECG signal, measured with the cardiac setting in LabScribe, underwent a 0.5 Hz high-pass filter and a 30 Hz low-pass filter.

For EMG signals, the signal-to-noise ratio was calculated as follows:

$$\text{SNR (dB)} = 10 \log \frac{Power_{signal}}{Power_{Noise}} = 20 \log \frac{Amplitude_{Signal(RMS)}}{Amplitude_{Noise(RMS)}} \quad (2)$$

## Fabrication of PILC printed 2D implantable devices

A poly(dimethylsiloxane) (PDMS; Sylgard 184, Dow Corning) pre-polymer solution was printed and utilized as an insulating substrate under in vivo conditions. After printing the PDMS prepolymer solution, it was cured at 80 °C for 30 min in an oven (OV3-30, JEIO TECH). PILC ink was directly printed on the as-cured PDMS substrate to fabricate implantable devices.

## Fabrication of PILC printed 3D implantable devices

A poly(dimethylsiloxane) (PDMS; Sylgard 184, Dow Corning) pre-polymer solution was printed and utilized as an insulating substrate under in vivo conditions. After printing the PDMS prepolymer solution, it was cured at 80 °C for 30 min in an oven (OV3-30, JEIO TECH). First, the PILC ink was directly 2D printed with high-resolution (50 μm) on the as-cured PDMS substrate to fabricate 3D implantable devices. Second, a 3D PILC microneedle with a high aspect ratio was directly printed onto 2D recording regions with a customized g-code. Finally, PDMS encapsulation was coated on the 3D implantable devices except for recording regions. The printed aspect ratio was calculated as seen in Supplementary Fig. 27.

## Material characterization

SEM images and EDS analyses were conducted using the SU8230 Scanning Electron Microscope (Hitachi High-Technologies Co., Japan). The PILC electrode was coated with a 3 nm layer of osmium to prevent surface charging before SEM measurements. Cryo-TEM images were acquired using the 200 kV Cryo-field emission TEM (Thermo Fisher, USA). These samples were prepared by diluting PILC ink in deionized water, followed by lyophilization in liquid nitrogen. FT-IR spectroscopy was performed using the Nicolet iS50 (Thermo Fisher Scientific Instrument, USA) while drying the PILC ink. X-ray Photoelectron Spectroscopy (XPS) was carried out with the sigma probe (Thermo VG Scientific, USA) for the PILC electrode. High-resolution Raman spectroscopy was conducted with ARAMIS (Horiba Jobin Yvon, France) using a 633 nm wavelength laser on the PILC electrode.

## Rheological characterization

The rheological characterization of the PILC inks was measured via a rheometer (Anton Paar MCR302, 8 mm parallel plate) at room temperature. The viscosity of PILC inks with varying concentrations was measured at a shear rate of 0.1 /s to 100 /s. The shear stress as a function of shear rate (0.01 /s to 100 /s) was measured under a fixed shear strain of 1% with different ratios of the PILC pellet and water. Based on the viscosity and yield stress at a shear rate of 0.01/s, the fluidic behavior of the PILC inks was determined. In addition, to prove the omnidirectional printability of the PILC inks, a strain sweep was conducted in oscillation mode at 0.01% to 100% shear strain at 6.28 rad/s angular frequency to monitor the storage modulus and loss modulus of the PILC inks.

## Mechanical characterization

The freestanding printed PILC electrodes were processed using a sharp razor blade to form a bar-type specimen with dimensions of 3 mm (width), 12 mm (length), and 7 μm (thickness). This specimen was secured between aluminum grips and attached with adhesives (DP-420 Off-White, 3 M, USA) at the water surface. Characterization at the water surface offers the benefits of hydration and minimizes deflection caused by gravity during the tensile test.

A load was applied to the swollen PILC electrode using a linear actuator (M-111.1 DG, PI, Germany) and a high-resolution load cell (LTS-5GA, KYOWA, Japan) integrated into linear XYZ stages. For precise measurement of strain data, a charge-coupled device camera was utilized in conjunction with a digital image correlation (DIC) system. In-situ stress-strain curves were derived from the collected load and strain data. Tensile tests were performed at room temperature to ensure the reliability and reproducibility of the obtained stress-strain data.

## Electrical characterization

The electrical conductivity of the PILC electrode was measured using a four-point probe (Keithley 2420, Keithley Instruments, USA). CV and EIS analyses were conducted utilizing a potentiostat (ZIVE SP1, Won A tech, Korea). Printed 2D and 3D PILC electrodes were used as both the working and counter electrodes, while gold electrodes were used as reference electrodes. PBS was used as the electrolyte. CV was measured at a scan rate of 300 mV/s from −0.6 to 0.6 V. CSC was calculated from the CV curves according to the equation as seen in Supplementary Fig. 23. Impedance of the printed PILC bioelectronic devices were evaluated every day for one month via a potentiostat (10 mV amplitude, 100 Hz to 1 MHz) in PBS solution to prove chemical stability.

## In vitro biocompatibility

In vitro biocompatibility was evaluated by incubation of PILC electrodes or control materials in cell media with NIH/3T3 cells (ATCC, CRL-1658) in a 5% $CO_2$ atmosphere at 37 °C for 24 hours. Cell media was composed of Dulbecco's Modified Eagle Medium cell culture medium, 10% fetal bovine serum and 1% penicillin/streptomycin. The NIH/3T3 cells were incubated in 24 well-cell culture dishes. The cell viability was measured with an MTT assay (ABCAM, ab228554) and absorbance at 460 nm was measured with a microplate reader (Molecular Devices, SpectraMax iD3). For cell viability images, LIVE/DEAD viability/cytotoxicity kit (ThermoFisher, L3224) was used and images were captured using a fluorescent microscope (Nikon Ti2, Japan). Live and dead cells were measured at excitation/emission spectra of 485 nm/530 nm and 530 nm/645 nm respectively.

## Experiments on human subjects

Ex-vivo experiments on e-skin electronics were performed under approval from the Institutional Review Board at Korea Advanced Institute of Science and Technology (protocol number: KH2023-153). We recruited healthy adults as participants (2 males; age = over 25 years old). All subjects voluntarily involved in experiments after informed consent.

## Animal experiments and surgical procedures

Male C57BL/6 Thy1-ChR2-YFP transgenic mice (8 weeks or older, Jackson Laboratory, strain B6.Cg-Tg(Thy1-COP4/EYFP)18Gfng/J, $n$ = 3) were utilized for in vivo recording of optically evoked electrocorticography. C57BL/6 N (8 weeks or older, Koatech, $n$ = 3) and C57BL/6 J (8 weeks or older, Jackson Laboratory, $n$ = 3) wild-type mice were utilized for in vivo sciatic nerve stimulation experiments, 2D and 3D endogenous electrocorticography signals and visually-evoked potentials. This study did not involve sex-based analysis. Mice were housed and maintained under conditions of 12 h light/dark cycle, 22–24 °C, 45% humidity and given ad libitum access to food and water. All mouse experiments were reviewed and approved by the KAIST Institutional Animal Care and Use Committee.

During in vivo experiments, mice were anesthetized with isoflurane inhalation (4 ~ 5% for induction, 1 ~ 2% for maintenance under $O_2$ flow). Vital signals such as respiratory rate and body temperature were monitored during surgery.

## In vivo sciatic nerve stimulation

After the application of anesthesia, the hair of C57BL/6 N (8 weeks or older, Koatech) or C57BL/6 J (8 weeks or older, Jackson Laboratory) wild-type mice was removed on the hindleg skin and the sciatic nerve was exposed by separating the gluteus maximus and bicep femoris muscles. A PILC device fabricated as described previously was placed under the sciatic nerve with two electrodes connected for electrical stimulation. An isolated pulse stimulator (Model 2100, AM Systems) was used to apply electrical stimulation (1 Hz, 5 ms pulse duration). A protractor was placed under the leg and video footage was recorded to measure the angle displaced by the leg during electrical stimulation.

## In vivo recording of optically evoked electrocorticography signals

After the application of anesthesia, male C57BL/6 Thy1-ChR2-YFP transgenic mice (8 weeks or older, Jackson Laboratory, strain B6.Cg-Tg(Thy1-COP4/EYFP)18Gfng/J) were located in a stereotaxic stage for a craniotomy. The skin on the scalp was cut to access the skull and a sterilized dental drill was used to expose a 3 mm × 3 mm cranial window. The 4-channel PILC device was placed on the cranial window and connected to a multichannel electrophysiology recording equipment (Lab Rat Ephys, Tucker-Davis Technologies) for recording of OEP signals. A blue laser (465 nm IOS-465 Intelligent Optogenetics System, RWD) was connected to a silica optical fiber for optical stimulation of the cortical surface. The optical fiber was centered at one of the four channels and optical stimulation was applied (1 Hz, 5 ms pulse duration).

For analysis, the OEP signals were analog filtered with a bandpass filter of 1 ~ 1000 Hz. After digital filtering low pass filtering (400 Hz), waveforms are aligned and segmented at each stimulation pulse onset and averaged in a custom code to visualize the average optically evoked electrocorticography signals.

## In vivo recording of 2D and 3D endogenous electrocorticography signals and visually-evoked potentials

Surgery was conducted in the same manner as the optically evoked electrocorticography signals on C57BL/6 N (8 weeks or older, Koatech) and C57BL/6 J (8 weeks or older, Jackson Laboratory) wild-type mice. A 2D or 3D PILC device was placed on top of the dura. During signal recording, the OEP signals are analog filtered with a bandpass filter of 3 ~ 300 Hz with multichannel electrophysiology recording equipment (Lab Rat Ephys, Tucker-Davis Technologies).

For the recording of visually-evoked potentials, a 2D or 3D PILC device was placed on top of the secondary visual cortex (−1 mm ML; −3 mm AP). A blue laser (465 nm IOS-465 Intelligent Optogenetics System, RWD) was utilized to apply optical stimulation from a distance to either the contralateral or ipsilateral eye. Waveforms were aligned at each stimulation pulse onset and averaged to produce average waveforms during visual stimulation using MATLAB.

## Statistical analysis

Origin Pro was utilized for statistical analysis. All groups were tested for normality with the Shapiro-Wilk normality test. Data tested normal for all distributions. The homogeneity of variance between sample groups was also tested with the F-test. A two-sample t-test was conducted between groups with equal variance to determine statistically different groups. If the variance between tested groups was not equal, a two-sample t-test with the Welch correction was conducted.

## Reporting summary

Further information on research design is available in the Nature Portfolio Reporting Summary linked to this article.

# Data availability

The authors declare that the data supporting the findings of this study are available within the article and its Supplementary Information files. Source data are provided in this paper.

# Code availability

The software code used in printing applications of PILC inks is available from the corresponding author upon request.

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

## Acknowledgements

This work was supported by the National Research Foundation (NRF) via the Creative Research Initiative Center (grant number: NRF-2021M3H4A1A03049075 and NRF-2022R1A2C2006076 to Steve.P) and Grant No. 2022-0-00020 (Imperceptible on-skin sensor devices for musculoskeletal monitoring and rehabilitation).

## Author contributions

B.O., S.B., and K.N. designed the study. S.B. devised the PILC ink design. B.O. and S.B. fabricated and characterized printed PILC electronics and bioelectronics. B.O. characterized entire printing condition for on-demand fabrication of bioelectronics with measurement of rheological properties. B.O. and K.N. facilitated and edited g-code for suspended structures. K.N. and Y.C. performed and characterized in-vitro cytotoxicity. K.N. and C.S. performed in vivo animal experiments. Y.S.L. facilitated the design and analysis of behavioral experiments for EMG and ECG recording. B.O. and M.S.J. conducted mechanical characterization. B.O., S.K. prepared PILC inks to demonstrate ex-vivo & in-vivo applications. Seongjun.P., Steve.P., T.S.K., S.M.P. supervised this work. B.O., S.B., K.N., Seongjun.P., Steve.P. wrote this paper. All of the authors discussed the results and reviewed this manuscript.

## Competing interests

The authors declare no competing interests.

## Additional information

[1]Department of Materials Science and Engineering, Korea Advanced Institute of Science and Technology (KAIST), 291 Daehak-ro, Yuseong-gu, Daejeon, Republic of Korea. [2]Department of Bio and Brain Engineering, Korea Advanced Institute of Science and Technology (KAIST), 291 Daehak-ro, Yuseong-gu, Daejeon, Republic of Korea. [3]Department of Convergence IT Engineering (CiTE), Pohang University of Science and Technology (POSTECH), 77 Cheongam-ro, Nam-gu, Pohang-si, Gyeongsangbuk-do, Republic of Korea. [4]Department of Mechanical Engineering, Korea Advanced Institute of Science and Technology (KAIST), 291 Daehak-ro, Yuseong-gu, Daejeon, Republic of Korea. [5]Department of Electrical Engineering, Pohang University of Science and Technology (POSTECH), 77 Cheongam-ro, Nam-gu, Pohang-si, Gyeongsangbuk-do, Republic of Korea. [6]Department of Mechanical Engineering, Pohang University of Science and Technology (POSTECH), 77 Cheongam-ro, Nam-gu, Pohang-si, Gyeongsangbuk-do, Republic of Korea. [7]Institute of Convergence Science, Yonsei University, Seoul, Republic of Korea. [8]KAIST Institute for NanoCentury, 291 Daehak-ro, Yuseong-gu, Daejeon, Republic of Korea. [9]These authors contributed equally: Byungkook Oh, Seunghyeok Baek, Kum Seok Nam. ✉e-mail: spark19@kaist.ac.kr; stevepark@kaist.ac.kr

