## [Peer Review File · Nature Communications]

3D Printable and Biocompatible PEDOT:PSS-Ionic Liquid Colloids with High Conductivity for Rapid On-demand Fabrication of 3D BioelectronicsREVIEWER COMMENTS

Reviewer #1 (Remarks to the Author):

The authors present 3D printable PEDOT:PSS-ionic liquid colloidal inks which have high electrical conductivity, storage modulus, and yield stress achieving outstanding self-supporting mechanical rigidity. This result is important to expand the possibility of biocompatible PEDOT:PSS printing in 3D structures overcoming conventional rheological properties of PEDOT:PSS inks (i.e., low yield stress and storage modulus). Also, the biocompatibility of the PILC ink via centrifugal removal of ionic liquids is presented by demonstrating the EMG arrays and implantable stimulators. The results are interesting and have impacts in the field of 3D-printed bioelectronics. So I would like to recommend publication in Nature Communications after minor revision.

1. The conductivity of 286 S/cm is impressive but why EMIM:TCB was used to prepare PILC? What are the key parameters to increase the conductivity?
2. In Fig. 2c, why strong shear thinning behavior of PILC is exhibited?
3. (minor comment) One of the bottlenecks in 3D printing is a time-consuming process during lifting-up of printed features. Is the omnidirectional printing video recorded in real-time?

Reviewer #2 (Remarks to the Author):

General Comment:

This work presents a one-shot strategy to prepare biocompatible and highly conductive PEDOT:PSS-ionic liquid colloidal (PILC) inks. The controllable rheological properties make PILC inks suitable for on-demand 3D printing of complex structures and bioelectronics. The topic is interesting, but the overall design and demonstration seems insufficient to prove the contribution and importance of this work. Some critical points are as follows:

Specific Comments:

Comment 1:

I am pretty concerned with the novelty of this work, cause the overall design and demonstration are too similar with previous recent publications such as Nat. Mater. 22, 895–902 (2023); Nat. Commun. 11, 1604 (2020); Adv. Mater. 35, 2304095 (2023); Adv. Funct. Mater. 2314471 (2023);, some of which have already been cited. The author should clearly define the novelty of this work by fully comparing with these reports.

Comment 2:

What do the authors mean of "limited to 2.5 dimensional patterning"? I can clearly see well-printed 3D structures in earlier works like 20 layer meshes, overhanging structures, cubes, and 3D rings by same DIW 3D printing in Nat. Commun. 11, 1604 (2020) & Small 2023, 2308778 (2023), and liquid-in-liquid DIW printing in Nat. Commun. 14, 4289 (2023). There are also some other reports showing similar capability of printing PEDOT:PSS into 3D structures. If these structures are defined as "limited to 2.5D", I don't think the authors have addressed such an issue, cause the mentioned PILC ink also contains a substantial amount of water.

Comment 3:

It seems that the main contribution of this work is increasing the conductivity of PEDOT:PSS. But the comparison of the results is unfair, cause the mentioned controls are mostly PEDOT:PSS hydrogels. The authors should compare their conductivity fairly by carefully measuring the water content of the materials. I believe that the water content should be much lower than typical PEDOT:PSS hydrogels since PSS chains have been removed substantially. BTW, the authors do have missed some important reports on PEDOT:PSS hydrogels showing very high electrical conductivity, such as Adv. Mater. 35, 2209324 (2023); Matter 5, 4407 (2022);

Comment 4:

Following the abovementioned comment, how is the mechanical property of the printed PILC

polymers? It seems that the author just reported the Young's modulus, without showing any evidence like strain-stress curve and other important parameters like tensile strength, toughness, and so on. Decent mechanical performance is also critical for applications the authors demonstrated like 3D circuit boards, epidermal electrodes, and implantable recording electrodes. I suggest the authors to supplement such experiments and compare their results with previous reports.

Comment 5:

In Fig. 4, the authors printed 3D conductive wires with overhanging features, which are still far from "3D circuits". Meanwhile, some literature demonstrated similar designs [ACS Appl. Mater. Interfaces 15, 57717 (2023)]. What is the main contribution of PILC polymers?

Comment 6:

Both PEDOT:PSS-based epidermal electrodes and ECoG recording electrodes have been previously reported with similar and even higher performances, as listed in previously mentioned publications. What's the novelty here? It seems that the parameters are not effectively improved compared to previous results, like SNR for epidermal electrodes, CSC and impedance. Also, what is the unique advantage or superior performance in these applications of increasing the conductivity of PEDOT:PSS? I can also see that the noise of the EMG signals increase probably due to the conductivity enhancement.

Comment 7:

A clear and detailed table is necessary for quantitative comparison, considering the figures about performance comparison are redundant and lack of unified literature database.

Comment 8:

How to calculate the "3D aspect ratio" should be clearly illustrated, such as which structure as well as detailed data on the height and width.

Comment 9:

Since applications in the last two figures are all printed mostly in a 2D manner by using a 3D printer, without demonstrating the new capability or functionality of well-printed 3D structures. More applications should be demonstrated to harness the advantages of 3D structures.

Comment 10:

Some figures are unclear and may confuse the readership. The schematic drawing in Fig. 1b and Fig. 1c needs a clear illustration between different particles and specific ink compositions. The Y-axis scale label of Fig. 2d may be better to be consistent with that of Fig. 2c. The abbreviation "3DP" in Fig. 5c and Fig. 6d requires clarification in the manuscript.

Reviewer #3 (Remarks to the Author):

This paper reports a new fabrication method for 3D printable, biocompatible, and highly conductive PEDOT:PSS-ionic liquid colloidal (PILC) inks that can be used in various bioelectronics applications, including the physiological detection (ECG and EMG) and in vivo optogenetic ECoG signal. Although this work shows significant outcomes, a few points should be clarified before publication.

Specific comments:

The author provides three major applications: omnidirectional printing, skin electrode printing, and in-vivo electrode printing. The mechanical study of the omnidirectional printed structure looks important to prove the ink's potential usage for 3D applications. However, the detailed experiments needed to be included.

For skin-electrode and in-vivo electrode applications, they look more like 2D electrode applications, which have a lot of comparable examples reported previously. Please clarify how this work is different from others.

How long does the ink drying time require for multilayer printing in Fig. 2h?

The skin contact impedance at a low-frequency level (~ 1 Hz) should be monitored for physiological signal monitoring.

What is the adhesion strength of skin electrodes?

In Supplementary Fig. 12, what happened to the electrode with additional strain? Maximum stretchable range? Repeatability?

In Fig. 6a, what is the bending radius?

There needs to be more description of why the performance in Fig. 6a-c is important for the in-vivo electrode application.

In Fig. 6e-f, the author said PILC ECoG device can record ECoG signals with high SNR and spatial selectivity. How do you validate this performance?

If applicable, please provide photo images of the electrode attached to the mouse's brain or nerves so that it can improve the manuscript quality.

Subject: Revised Manuscript, “3D Printable and Biocompatible PEDOT:PSS-Ionic Liquid Colloids with High Conductivity for Rapid On-demand Fabrication of 3D Bioelectronics”

Reviewer’s comments are *italicized*, authors’ responses are in blue, and text quotes from the original and revised manuscript are in red.

Reviewer #1

Overall comment: *The authors present 3D printable PEDOT:PSS-ionic liquid colloidal inks which have high electrical conductivity, storage modulus, and yield stress achieving outstanding self-supporting mechanical rigidity. This result is important to expand the possibility of biocompatible PEDOT:PSS printing in 3D structures overcoming conventional rheological properties of PEDOT:PSS inks (i.e., low yield stress and storage modulus). Also, the biocompatibility of the PILC ink via centrifugal removal of ionic liquids is presented by demonstrating the EMG arrays and implantable stimulators. The results are interesting and have impacts in the field of 3D-printed bioelectronics. So I would like to recommend publication in Nature Communications after minor revision.*

We thank the reviewer for recognizing the quality and strength of our work and appreciate their valuable time for giving helpful comments.

1. *The conductivity of 286 S/cm is impressive but why EMIM:TCB was used to prepare PILC? What are the key parameters to increase the conductivity?*

We explored other commonly used ionic liquids (EMIM:ES and HMIM:TCB) to demonstrate if they result in similar conductivity as EMIM:TCB. However, we found that printable inks were not formed when using EMIM:ES and HMIM:TCB after centrifugation (**Fig. R1**). Therefore, direct comparison of the conductivity of PEDOT:PSS with EMIM:ES and HMIM:TCB cannot be made with PILC inks.

Due to the experimental difficulty in providing a direct explanation, we discuss theoretically why we chose EMIM:TCB. The key parameter for increasing the conductivity of the PEDOT:PSS-ionic composite is the ionic exchange between PEDOT:PSS and ionic liquids. According to a study published in J. Am. Chem. Soc. 140, 5375–5384 (2018), EMIM:TCB has one of the lowest Gibbs free energy among ionic liquids when reacting with PEDOT:PSS. The low Gibbs free energy of EMIM:TCB facilitates more effective ion exchange with PEDOT:PSS. This enhanced and spontaneous ion exchange promotes the formation of interconnected PEDOT chains, thereby increasing the conductivity of the PEDOT:PSS-ionic liquid composite.

Fig. R1 (Supplementary Figure 5 in the revised manuscript) | Centrifuge effects of PEDOT:PSS and other ionic liquid (EMIM:ES and HMIM:TCB).

(Revised Main Text, Page 5-6, Line 21)

By adding EMIM:TCB during ink fabrication, PEDOT:PSS demonstrated phase separation after centrifugation (**Supplementary Fig. 4**). In contrast, when no ionic liquid is used or DMSO is utilized, no phase separation is observed, demonstrating the key role of the ionic liquid in ink formation. Furthermore, adding other ionic liquids (1-Ethyl-3-methylimidazolium ethyl sulfate (EMIM:ES) and 1-hexyl-3-methylimidazolium tetracyanoborate (HMIM:TCB)) during ink fabrication, PEDOT:PSS did not demonstrate phase separation after centrifugation (**Supplementary Fig. 5**). To test the hypothesis on ionic liquid facilitated-hydrogen bonding, the FT-IR spectrum was measured as a function of PILC ink drying time (thus at different amounts of solvent in the film) (**Fig. 2b**). As drying time increases, the –OH vibration peak progressively shifted to a lower wavenumber, indicating that the hydrogen bonded network is increasing.^{26,27}

2. In Fig. 2c, why strong shear thinning behavior of PILC is exhibited?

Shear thinning occurs in the presence of temporary networks and structures in the fluid, which gets broken down with increasing shear rate, hence resulting in decreased viscosity. Such a network is formed by hydrogen bonding, as mentioned in the manuscript. To demonstrate the effect of this hydrogen bonding, urea, a substance commonly utilized to disrupt of hydrogen bonding¹, was mixed with PILC inks. Since urea disrupts hydrogen bonding between interconnected PEDOT chains, the viscosity and shear stress of PILC inks decrease with increased molar ratios of urea (**Fig. R2**).

References

[1]: Hua, L., Zhou, R., Thirumalai, D. & Berne, B. J. Urea denaturation by stronger dispersion interactions with proteins than water implies a 2-stage unfolding. *Proceedings of the National Academy of Sciences* 105, 16928–16933 (2008).

Fig. R2 (Supplementary Figure 7 in the revised manuscript) | Shear-thinning behavior of PILC inks at various Urea concentration (0.5 M and 2.0M).

(Revised Main Text, Page 6, Line 14)

The pristine PEDOT:PSS ink has low viscosity at all shear rates, which renders it non-printable. In contrast, PILC inks shows shear thinning behavior with high viscosity at low shear rates **by enhancing hydrogen bonding between PEDOT:PSS-ionic liquid colloidal particles**, along with long term ink stability (**Supplementary Fig. 6**). In addition, the pristine ink shows Newtonian behavior; whereas the PILC ink exhibits rheopectic behavior, with a high shear stress at low shear rates, demonstrating why PILC ink can maintain its structure after printing.²⁸ **To analyze that PILC inks shows higher shear thinning effect than pristine PEDOT:PSS inks, urea that has been used for disruption of hydrogen bonding²⁹ was introduced in PILC inks. The shear thinning effects of PILC inks decreased with increasing urea molar ratio since it disrupts hydrogen bonding between interconnected PEDOT chains (**Supplementary Fig. 7**).**

3. *One of the bottlenecks in 3D printing is a time-consuming process during lifting-up of printed features. Is the omnidirectional printing video recorded in real-time?*

We thank the reviewer for this comment. Yes, it was recorded in real-time.

We thank the reviewer for their insightful comments and the time spent for their thorough review of our work. We have worked to fully address these comments with additional data that have substantially improved the manuscript.

Reviewer #2

Overall comment: *This work presents a one-shot strategy to prepare biocompatible and highly conductive PEDOT:PSS-ionic liquid colloidal (PILC) inks. The controllable rheological properties make PILC inks suitable for on-demand 3D printing of complex structures and bioelectronics. The topic is interesting, but the overall design and demonstration seems insufficient to prove the contribution and importance of this work. Some critical points are as follows:*

We thank the reviewer for the appraisal of our work and appreciate his/her valuable time to give helpful comments and suggestions. Below, we address the reviewer's concern on a point-by-point basis to convey that this work indeed has the novelty and significance for publication in Nature Communications.

Comment 1: I am pretty concerned with the novelty of this work, cause the overall design and demonstration are too similar with previous recent publications such as Nat. Mater. 22, 895–902 (2023); Nat. Commun. 11, 1604 (2020); Adv. Mater. 35, 2304095 (2023); Adv. Funct. Mater. 2314471 (2023);, some of which have already been cited. The author should clearly define the novelty of this work by fully comparing with these reports.

We sincerely appreciate the reviewer's feedback and valuable comments. In comments 1, 2, 5, and 9, the reviewer raised concerns regarding the novelty of the PILC ink and its demonstrations. To avoid repetition in our response, we will first address this point below.

We respectfully disagree that the design of the PILC ink is similar to recent publications, as it is the first demonstration of a general purpose PEDOT:PSS ink with a unique set of properties and versatile applicability. Specifically, we distinctively present a one-shot strategy that result in immediate usability of printed PILC electrodes without complex post-treatment, along with a combination of unique properties such as biocompatibility, high conductivity, and three-dimensional printability (**Fig. R1**). A table comparing our ink properties to that of previous PEDOT:PSS inks (including the papers mentioned by the reviewer) is presented in **Table R1 and Table R2.**

Regarding the demonstrations in our original manuscript, we agree that certain demonstrations are similar with that of previous publications. To make a clear differentiation, we have added in-vivo 3D bioelectronic device demonstration for neural probes (**Fig. R2**), which to the best of our knowledge, has not been demonstrated previously using PEDOT:PSS. We also demonstrated the ability of the PILC ink to create a suspended PILC artificial 3D tissue structure (diagonal overhang: ~ 5 mm with various printing angle) (**Fig. R3**). This potentially enables 3D organoid-on-a-chip for organ-specific genetic disease monitoring system, where the PILC can bridge two different interfaces with a high vertical and horizontal gap. We also made other complex 3D structures with different aspect ratios (**Fig. R4**). Finally, a table comparing the printing versatility of our ink to that of other inks is presented in **Table R2.**

Fig. R1 (Supplementary Fig. 3 in the revised manuscript) | Schematic illustration of versatile applicability of PILC inks from 2D *In-vivo* / *Ex-vivo* electrode and 3D diagonally printed circuit lines to 3D *In-vivo* electrodes.

(Revised Main Text, Page 4, Line 20)

The combination of unique material properties (high conductivity, high printing resolution, biocompatibility before post processing, and high 3D aspect ratio) exhibits the broad applicability of the PILC ink as a general purpose PEDOT:PSS ink in bioelectronic applications **(Fig. 1e-f, Supplementary Fig. 3 and Supplementary Table. 1-Table 3)**.

Ref	Viscosity (Pa*s)	Yield stress (Pa)	Storage modulus (Pa)	Conductivity (S/cm)	Intrinsic biocompatibility	3D-Printed Layers (aspect ratio)	Water content (%)	Printing resolution (um)
[1]	5×10^2	10^2	5×10^3	28 (dry) ----- 155 (wet)	X (>24 hr post annealing)	20 (N/R)	87	30
[2]	23	N/R	N/R	0.022	N/A (> 1.5 hr post-treatment)	N/A	260	94
[3]	10^2	10^4	5×10^3	858.1 (Post-acid treatment)	N/A	25 (N/R)	N/R	50
[4]	3×10^3	4×10^4	4×10^3	72 (wet) ----- 654 (Post-acid treatment)	N/A	N/R (8~15)	N/R	15
[5]	100	N/R	N/R	11	N/A (> 24 hr post-treatment)	N/A	80	100
[6]	15	N/R	10^3	0.0159	N/A (> 1 hr post-treatment)	N/A	N/R	150
[7]	10^3	10^3	330	1,200 (Post-acid treatment)	N/A	10 (N/R)	N/R	50
[8]	10^2	10^3	10^3	0.09	N/A (> 24 hr post-treatment)	N/A	75	80
This Work	10^6	10^3	10^5	286	O	>100 (>4)	28.75	50

N/A : Not Applicable
N/R : Not Reported

Table R1. (Supplementary Table. 1 in the revised manuscript) | Comparison with previous papers using printable PEDOT:PSS inks

Ref	Method	Resolution (um)	2D In / Ex-vivo Application	3D Suspended Structure	3D In-vivo Application
[1]	Nozzle printing	30	O (In-vivo)	X	X
[2]	Liquid-in-Liquid Printing	94	X	O	X
[4]	Nozzle printing	15	X	O	X
[5]	Nozzle printing	100	O (In-vivo)	X	X
[6]	Nozzle printing	150	O (Ex-vivo)	X	X
[8]	Nozzle printing	80	O (In-vivo)	X	X
[10]	Molding	N/R	O (In-vivo)	X	X
[11]	Laser-Induced phase separation	6	O (In-vivo)	X	X
[12]	orthogonal photochemistry printing	100	O (Ex-vivo)	O	X
[13]	Coating	N/R	O (In-vivo)	X	X
[14]	Laser cutting	150	O (In-vivo)	O	X
This Work	Nozzle printing	50	O (In-vivo & Ex-vivo)	O	O

N/R : Not Reported

Table R2. (Supplementary Table. 3 in the revised manuscript) | Comparison of versatility with previous papers using PEDOT:PSS inks

(Revised Main Text, Page 4, Line 17)

The combination of unique material properties (high conductivity, high printing resolution, biocompatibility before post processing, and high 3D aspect ratio) exhibits the broad applicability of the PILC ink as a general purpose PEDOT:PSS ink in bioelectronic applications (**Fig. 1e-f, Supplementary Fig. 3 and Supplementary Table. 1-Table 3**).

Fig. R2 (Fig. 7 in the revised manuscript) | 3D PILC devices for flexible in vivo bioelectronics **a**, Schematic of 3D PILC arrays for fabrication of flexible, biocompatible 3D devices. 3D PILC arrays enable customizable interfacing with neural signals from deeper cortical layers and hippocampus, which are difficult to achieve with 2D surface arrays. **b**, Image of 3D PILC arrays enabled through high aspect ratio printing of PILC ink **c**, Power spectral density (PSD) of 3D and 2D PILC arrays for recording of characteristics endogenous neural signals from deeper brain regions (i.e., hippocampus). The red shaded area indicates the frequency band of theta oscillations (4~12 Hz). The theta band peaks and their harmonic frequencies in the PSD are marked with red arrows. **d**, Spectrogram of 3D (top) and 2D(bottom) PILC arrays. **e**, Raw local field potentials recorded from 3D PILC arrays (top) and 2D PILC arrays (bottom). **f**, Schematic of sensory stimulation of the secondary visual cortex in an in vivo mouse model. **g-i**, Evoked neural signals from a (g) 2D array during contralateral stimulation, (h) 3D array during contralateral stimulation, and (i) 3D array during ipsilateral stimulation.

(Revised Main Text, Page 12, Line 1)

We also demonstrated that the versatile printing of the PILC ink enables the fabrication of 3D bioelectronic devices, such as vertical spike arrays (**Fig. 7a-b**). These flexible and biocompatible devices consisting of array of PILC electrodes with high aspect ratio

enables the recording of neural signals from deep brain regions, such as the hippocampus, which is difficult to achieve with 2D surface arrays. We validate that the recorded signals demonstrate characteristics of hippocampal signals by examining the signal frequency spectrum. A dominant signal characteristic of the hippocampus are theta wave oscillations, which are signals prominent in the 4~8Hz frequency band³⁶. The 3D PILC arrays demonstrates peaks at ~5 Hz and its harmonic frequencies as shown in the power spectral density (**Fig. 7c**), spectrogram (**Fig. 7d**), and raw signal waveforms (**Fig. 7e**). While in contrast, this peak is not observed in the 2D PILC arrays, due to their spatial restriction to the surface of the brain (**Fig. 7c-e**).

In addition, the 3D PILC arrays demonstrate high fidelity recording during electrophysiology of sensory evoked potentials (**Fig. 7f**). Cortical neural signals demonstrate maximal amplitude when neural probes are inserted 300~500µm below the cortical surface (i.e., Layer IV~V)³⁷. We target the secondary visual cortex, which has been shown to have reduced signal amplitude in comparison to the primary visual cortex upon application of visual stimuli.

Upon application of visual stimulus to the eye contralateral to the PILC array, signals-correlated to stimulus intensity were observed in the 3D PILC array in contrast to the 2D array (**Fig. 7g-h**). The correlation in visually-evoked signal amplitude and stimulus intensity verifies successful recording of sensory evoked potentials. In addition, to verify that the signal is not a recording artifact, ipsilateral visual stimulation is also conducted. No signal is observed upon ipsilateral stimulation, suggesting there is minimal light-induced artifacts (**Fig. 7i**).

References

- 36** Nuñez, Angel, and Washington Buño. "The theta rhythm of the hippocampus: from neuronal and circuit mechanisms to behavior." *Front. cell. neurosci.* **15**, 649262 (2021).
- 37** Narcisse, Darryl, et al. "Monitoring visual cortical activities during progressive retinal degeneration using functional bioluminescence imaging." *Front. neurol.* **15**, 750684 (2021).

(Revised Main Text, Page 19, Line 11)

In vivo recording of 2D and 3D endogenous electrocorticography signals and visually-evoked potentials

Surgery was conducted in the same manner as the optically evoked electrocorticography signals. A 2D or 3D PILC device was placed on top of the dura. During signal recording, the OEP signals are analog filtered with a bandpass filter of 3~300 Hz with a multichannel electrophysiology recording equipment (Lab Rat Ephys, Tucker-Davis Technologies).

For recording of visually-evoked potentials, a 2D or 3D PILC device was placed on top of secondary visual cortex (-1 mm ML; -3 mm AP). A blue laser (465nm IOS-465 Intelligent Optogenetics System, RWD) was utilized to given direct optical stimulation from a ~5cm distance to either the contralateral or ipsilateral eye. Waveforms were aligned at each stimulation pulse onset and averaged to produce average waveforms during visual stimulation using MATLAB.

Fig. R3 (Supplementary Fig. 14 in the revised manuscript) | 3D diagonally printed circuits lines for achieving an artificial 3D tissue with artificial kidney made of Ecoflex-0020

(Revised Main Text, Page 9, Line 1)

The PILC ink can be utilized to print suspended interconnects by simultaneously moving the x- and z- stage in the air (**Fig. 4b-c**). In addition, 3D diagonally suspended circuit lines can be printed onto an artificial kidney (made of Ecoflex-0020, with a height of 7 mm), demonstrating the potential of employing a 3D interconnection of PILC inks via omnidirectional printing. This approach can effectively bridge the two different interfaces, even with a high angle and high gap, (as shown in **Supplementary Fig. 14**).

(Revised Main Text, Page 15, Line 5)

2D & 3D printing of PILC inks

A direct-ink-writing printer (BIO X6, CELLINK) was used to print the PILC inks onto various films. The as-prepared PILC ink was loaded into a 3mL syringe and was utilized for the pressure-driven extrusion of PILC inks during printing. The tip diameter of the nozzles and printing speeds were determined in accordance with the specified requirements (nozzle diameter: 30 μm to 250 μm , printing speed: 6.4 mm/s to 12 mm/s). For the 3D printing of PILC inks, such as omnidirectional printing, MATLAB software (R2019a, The MathWorks, Inc.) was utilized to customize the g-code and incorporated automatic x-stage movements (printing speed: 1.4 mm/s to 2.0mm/s with 30 Gauge nozzles), z-stage movements (printing speed: 1.4 mm/s to 2.0mm/s) and pressure control (50 - 100 kPa).

Fig. R4 (Fig. 5 in the revised manuscript) | 3D high aspect ratio printing of PILC inks with air printing w/o supported layers. a, Images of 3D high-aspect ratio printing of PILC inks. **b,** Images of air printing of PILC inks without supported layers. **c,** Images of 3D printed PILC structure that mimics pyramid structures. **d,** Images of an LED mounted on the 3D printed PILC structure.

(Revised Main Text, Page 9, Line 12)

The improved yield stress and storage modulus of the PILC ink, compared to pristine PEDOT:PSS inks and other PEDOT:PSS composites, enable the 3D high aspect ratio printing of PILC inks without any collapse of printed structures during printing (**Fig. 5a**). Additionally, due to its high yield stress and storage modulus, the PILC ink can be printed in air without any support layers, enabling the creation of bridges with the potential for 3D interconnections of individual circuits (**Fig. 5b**). Leveraging these capabilities for 3D high aspect ratio and air printability, an LED chip was mounted on a pyramid structure created using PILC inks after printing (**Fig. 5c**) and subsequently operated (**Fig. 5d**).

(Revised Main Text, Page 15, Line 16)

Fabrication of a 3D high aspect ratio circuit for operation an LED chip

To demonstrate the fabrication of a 3D high aspect ratio circuit, a SMD LED chip (2.6 V; HSMH-H170 with a 7 mm of printed height) was mounted on a pyramid PILC structure. The PILC ink was connected onto both the cathode and anode with silver paste and connected to Teflon-wrapping wires (SME). To turn on the LED, a voltage of 2.6 V was applied to the LED chip through a source meter (Keithley 2400, Tektronix Inc.).

Comment 2: What do the authors mean of “limited to 2.5 dimensional patterning”? I can clearly see well-printed 3D structures in earlier works like 20 layer meshes, overhanging structures, cubes, and 3D rings by same DIW 3D printing in Nat. Commun. 11, 1604 (2020) & Small 2023, 2308778 (2023), and liquid-in-liquid DIW printing in Nat. Commun. 14, 4289 (2023). There are also some other reports showing similar capability of printing PEDOT:PSS into 3D structures. If these structures are defined as “limited to 2.5D”, I do not think the authors have addressed such an issue, cause the mentioned PILC ink also contains a substantial amount of water.

We appreciate the reviewer’s valuable feedback. The PILC enables versatile application for tall 3D structures (> 100 printed layers compared to that of previous reported ~30 printed layers¹), high-resolution 2D structures, and improved 3D suspended structures with diagonally overhanging 3D circuit lines that (diagonal overhang: ~5-6 mm with various printing angle) exceeds previous work^{1,2}.

In addition, contrary to our work, in previous studies, time-consuming post-treatment was need to address cytotoxicity¹ and polymerization³ after printing PEDOT:PSS inks. Furthermore, liquid in liquid printing³ is not suitable for integration with 3D circuitry due to the incompatibility of immersing electronics in water.

We have removed the phrase "limited to 2.5-dimensional patterning" to eliminate any ambiguity.

Reference

[1]: Yuk, H. et al. 3D printing of conducting polymers. Nat Commun 11, 1604 (2020).

[2]: Yu, J. et al. 3D Printing of Robust High-Performance Conducting Polymer Hydrogel-Based Electrical Bioadhesive Interface for Soft Bioelectronics. Small, 2308778.

[3]: Xie, X. et al. Liquid-in-liquid printing of 3D and mechanically tunable conductive hydrogels. Nat Commun 14, 4289 (2023).

(Revised Main Text, Page 2, Line 6)

However, previously developed PEDOT:PSS inks have not been able to fully utilize the advantages of commercial 3D printing due to its long post treatment times, **difficulty in high aspect ratio printing**, and low conductivity.

Comment 3: It seems that the main contribution of this work is increasing the conductivity of PEDOT:PSS. But the comparison of the results is unfair, cause the mentioned controls are mostly PEDOT:PSS hydrogels. The authors should compare their conductivity fairly by carefully measuring the water content of the materials. I believe that the water content should be much lower than typical PEDOT:PSS hydrogels since PSS chains have been removed substantially. BTW, the authors do have missed some important reports on PEDOT:PSS hydrogels showing very high electrical conductivity, such as Adv. Mater. 35, 2209324 (2023); Matter 5, 4407 (2022);

Thank you for your insightful comment. **Fig. 1e** and **1f** was intended to describe the unique characteristics of PILC ink compared to other previous reported “3D printable” PEDOT:PSS inks. In order to eliminate ambiguity, we have revised the captions of **Fig. 1e** and **1f**.

Additionally, we have included a table to compare the water contents, resolution, biocompatibility, and conductivity of our PILC ink with other 3D printable PEDOT:PSS inks (**Table R1** and **Table R2**). We hope these revisions address your concerns adequately and contribute to a clearer understanding of our findings.

(Revised Main Text, Page 21)

Fig. 1 | PEDOT:PSS ionic liquid colloidal (PILC) ink for 3D printed bioelectronics **a**, Schematic illustration (left) and SEM image (right) of the PILC ink enabling 3D printed structures with high structural integrity through the dense packing of colloidal particles. **b**, Schematic illustrations of the PILC ink with minimal ionic liquid content and good structural integrity. **c**, Schematic illustrations of conventional PEDOT:PSS ionic liquid composites with cytotoxic ionic liquid components and poor structural integrity. **d**, Schematic illustration of the PILC ink for rapid prototyping of PILC bioelectronics. **e**, Conductivity, and printing resolution of the PILC ink in comparison to previous works **using 3D printable PEDOT:PSS inks**. **f**, 3D aspect ratio, conductivity and biocompatibility of the PILC ink before post treatment in comparison to previous works **using 3D printable PEDOT:PSS inks**.

Comment 4: Following the abovementioned comment, how is the mechanical property of the printed PILC polymers? It seems that the author just reported the Young's modulus, without showing any evidence like strain-stress curve and other important parameters like tensile strength, toughness, and so on. Decent mechanical performance is also critical for applications the authors demonstrated like 3D circuit boards, epidermal electrodes, and implantable recording electrodes. I suggest the authors to supplement such experiments and compare their results with previous reports.

We thank the reviewer for mentioning this point. Additional mechanical characterization (tensile strength, toughness, maximum strain range, repeatability and adhesion strength) was added to the revised manuscript (**Fig. R5**). We have also determined that in contrast to freestanding PILC films, which can be stretched up to 10% strain, printed PILC electrodes on a stretchable substrate (PDMS) showed 90% stretchability and cyclability at 70% strain for 1,000 cycles (**Fig. R6, R7**). Such an effect can be attributed to strain energy dissipation as previously reported^{1,2}.

Moreover, adhesion strength was measured through the attachment of printed PILC film between two wet porcine skins. The results show that an adhesive strength of 1.884 N/mm is

need to peel off the two porcine skins (**Fig. R8**). A table comparing the mechanical properties of our film to that of other works has been added to the revised manuscript (**Table R3**).

Reference

[1]: Yang, J. C. et al. Geometrically engineered rigid island array for stretchable electronics capable of withstanding various deformation modes. *Sci. Adv*, 8, eabn3863 (2022).

[2]: Cai, M., Nie, S., Du, Y., Wang, C. & Song, J. Soft Elastomers with Programmable Stiffness as Strain-Isolating Substrates for Stretchable Electronics. *ACS Appl. Mater. Interfaces*, 11, 14340–14346 (2019).

Fig. R5 (Supplementary Fig. 16 in the revised manuscript) | Mechanical properties of freestanding PILC films. **a**, A photograph of the set-up used to measure the mechanical properties of the PILC ink on water. **b**, Stress versus strain of the PILC film. Printed PILC films demonstrate low Young's modulus (750 kPa) similar to tissues.

(Revised Main Text, Page 9, Line 20)

The unique properties of the PILC ink also makes it an ideal material for rapid on-demand fabrication of on-skin bioelectronics for health monitoring applications. As electrodes printed from PILC ink have similar mechanical properties (Young's modulus 750 kPa, **Tensile Strength 139.3 kPa, and Toughness 4.0 kJ/m³** as seen in **Supplementary Fig. 16**) with biological tissues,³³ PILC electrodes can be utilized as soft electrodes to interface with tissue.

Fig. R6 (Supplementary Fig. 17 in the revised manuscript) | Mechanical properties of printed PILC on a PDMS substrate at strain rate: 50 mm/min.

Fig. R7 (Supplementary Fig. 18 in the revised manuscript) | Repeatability of printed PILC on a PDMS substrate for 1,000 cycles @ 70% strain at strain rate: 50 mm/min).

(Revised Main Text, Page 10, Line 2)

Furthermore, printed PILC electrodes on soft substrate (PDMS) have stretchability up to 90% strain and cyclability at 70 % strain for 1,000 cycles (**Supplementary Fig. 17 and Fig. 18**) in contrast to freestanding PILC film, which can only be stretched up to 10% strain. This can be attributed to energy dissipation mechanism as previously reported^{34,35}.

References

34. Yang, J. C. et al. Geometrically engineered rigid island array for stretchable electronics capable of withstanding various deformation modes. *Science Advances* 8, eabn3863 (2022).
35. Cai, M., Nie, S., Du, Y., Wang, C. & Song, J. Soft Elastomers with Programmable Stiffness as Strain-Isolating Substrates for Stretchable Electronics. *ACS Appl. Mater. Interfaces* 11, 14340–14346 (2019).

Fig. R8 (Supplementary Fig. 21 in the revised manuscript) | Adhesion property of PILC between wet porcine skins.

(Revised Main Text, Page 10, Line 6)

After printing the PILC ink on a super hydrophobic (polypropylene) flexible substrate, the PILC electrodes can be transferred on to wet skin under gentle pressure (**Supplementary Fig. 19a, Supplementary Fig. 20**) with adhesion strength of 1.884 N/mm between wet porcine skins and printed PILC films, which is need to peel off wet porcine skins attached with film of printed PILC (**Supplementary Fig. 21**).

Ref	Method	Young's modulus (kPa)	Tensile strength (MPa)	Stretchability (%)	Charge Storage Capacity (mC cm ⁻²)	Initial Conductivity (S/cm)
[1]	Nozzle printing	1,100	N/R	N/R	N/R	28 (dry) 155 (wet)
[2]	Liquid-in-Liquid Printing	41.2 - 3,875	1 - 2.5	76 - 800	N/R	0.022
[5]	Nozzle printing	1,000	0.5 - 3.5	400	6	11
[6]	Nozzle printing	5 - 65	110	349	12.37	0.0159
[8]	Nozzle printing	650	0.9	120	5.83	0.09
[9]	Photolithography	2,000	0.5 - 2.2	15 - 35	60	20-40
[10]	Molding	25	130	610	80	247
[11]	Laser-Induced phase separation	120	15	15	32.13	670
This Work	Nozzle printing	750	0.139	88.6	2D : 47.04 3D : 104.827	286

N/R : Not Reported

Table R3. (Supplementary Table. 2 in the revised manuscript) | Comparison with previous papers using PEDOT:PSS inks

(Revised Main Text, Page 4, Line 17)

The combination of unique material properties (high conductivity, high printing resolution, biocompatibility before post processing, and high 3D aspect ratio) exhibits the broad applicability of the PILC ink as a general purpose PEDOT:PSS ink in bioelectronic applications (Fig. 1e-f, Supplementary Fig. 3 and Supplementary Table. 1-Table 3).

Comment 5: In Fig. 4, the authors printed 3D conductive wires with overhanging features, which are still far from “3D circuits”. Meanwhile, some literature demonstrated similar designs [ACS Appl. Mater. Interfaces 15, 57717 (2023)]. What is the main contribution of PILC polymers?

We appreciate your valuable comment. While [ACS Appl. Mater. Interfaces 15, 57717 (2023)] presents designs with 3D arrays, it is worth noting that this material may not be directly applicable to bioelectronic applications due to its use of post-acid treatment. As mentioned above, to emphasize our novelty and significance, several in-vivo bioelectronic demonstrations were newly added to the revised manuscript.

Comment 6: Both PEDOT:PSS-based epidermal electrodes and ECoG recording electrodes have been previously reported with similar and even higher performances, as listed in previously mentioned publications. What’s the novelty here? It seems that the parameters are not effectively improved compared to previous results, like SNR for epidermal electrodes, CSC and impedance. Also, what is the unique advantage or superior performance in these applications of increasing the conductivity of PEDOT:PSS? I can also see that the noise of the EMG signals increase probably due to the conductivity enhancement.

We sincerely appreciate your insightful comment. Previously reported materials used in PEDOT:PSS bioelectronics often require lengthy cytotoxic material removal processes. In contrast, our PILC ink offers intrinsic biocompatibility, enabling versatile and rapid fabrication of electronics. Moreover, the excellent rheological properties of the PILC ink, with a storage modulus of approximately 10^5 Pa and yield stress of approximately 10^3 Pa, allow for the creation of high-aspect ratio PILC bioelectronics.

We acknowledge that the application of PILC electrode towards epidermal electronics may not effectively showcase the merits of the PILC ink. Therefore, this demonstration was moved to supplementary information (**Fig. R9**). Instead, we have added 3D bioelectronics demonstrations as a figure in the main text, and we have added **Table R2** to highlight the versatile printability of our ink. We believe that these changes will further underscore the unique capabilities of our PILC ink.

Fig. R9 (Supplementary Fig. 19 in the revised manuscript) | 3D printed PILC electrodes for on-skin bioelectronics. a, Images of multi-electrode EMG array printed with the PILC ink (left) and subsequent transfer to wet skin (right). **b**, Impedance spectra of PILC EMG electrodes in comparison to commercial 3M electrodes. **c**, Schematic (left) and average ECG waveform of the PILC ECG electrodes (middle) and conventional 3M electrodes (right). **d**, Schematic (left) and EMG waveform (right) collected with PILC EMG electrodes during bicep contraction.

(Revised Main Text, Page 10, Line 5)

We demonstrate the use of PILC ink to fabricate a large area on-skin e-tattoo (**Supplementary Fig. 19a**). After printing the PILC ink on a super hydrophobic (polypyrene) flexible substrate, the PILC electrodes can be transferred on to wet skin under gentle pressure (**Supplementary Fig. 19a**, **Supplementary Fig. 20**) with adhesion strength of 1.884 N/mm between wet porcine skins and printed PILC films, which is need to peel off wet porcine skins attached with film of printed PILC (**Supplementary Fig. 21**). The high conductivity and conformal contact of PILC electrodes results in lower interfacial skin impedance (200 kΩ·mm² at 1 kHz) compared to that of the commercial 3M electrodes (8,000 kΩ·mm² at 1 kHz) (**Supplementary Fig. 19b**).

Utilizing the lower interfacial skin impedance, PILC electrodes demonstrate high quality recordings of both ECG (electrocardiogram) and EMG (electromyogram) waveforms. PILC electrodes demonstrated an increased signal amplitude in the ECG waveform in comparison with commercial 3M electrodes (**Supplementary Fig. 19c**). In addition, PILC electrodes demonstrate high signal to noise ratios (16.80 dB) during EMG recordings of bicep muscles (**Supplementary Fig. 19d**).

Comment 7: A clear and detailed table is necessary for quantitative comparison, considering the figures about performance comparison are redundant and lack of unified literature database.

We sincerely appreciate the insightful suggestion to improve this work. We have added three supplementary tables to the revised manuscript (**Table R1-R3**) for quantitative comparison.

Supplementary Table 1 compares the key material properties for 3D printable bioelectronic applications, such as the rheological properties, conductivity, intrinsic biocompatibility, and resolution, amongst various 3D printable PEDOT:PSS inks. Supplementary Table 2 provides key intrinsic material properties of the PILC ink, such as mechanical properties, charge storage capacity, and initial conductivity in comparison to previously reported PEDOT:PSS inks. Finally, Supplementary Table 3 compares the printing versatility of various PEDOT:PSS inks with those of the PILC ink.

These supplementary tables aim to provide a comprehensive and unified database for readers to easily access and compare the performance of different PEDOT:PSS inks. We believe that these additions will enhance the clarity and distinctiveness of our work.

(Revised Main Text, Page 4, Line 17)

The combination of unique material properties (high conductivity, high printing resolution, biocompatibility before post processing, and high 3D aspect ratio) exhibits the broad applicability of the PILC ink as a general purpose PEDOT:PSS ink in bioelectronic applications (**Fig. 1e-f, Supplementary Fig. 3 and Supplementary Table. 1-Table 3**).

(Revised Supplementary Information, Page 32)

Reference

1. Yuk, H. et al. 3D printing of conducting polymers. *Nat Commun* **11**, 1604 (2020).
2. Xie, X. et al. Liquid-in-liquid printing of 3D and mechanically tunable conductive hydrogels. *Nat Commun* **14**, 4289 (2023).
3. Ghaderi, S., Hosseini, H., Arash Haddadi, S., Kamkar, M. & Arjmand, M. 3D printing of solvent-treated PEDOT:PSS inks for electromagnetic interference shielding. *J. Mater. Chem. A*. **11**, 16027–16038 (2023).
4. Xing, W. et al. Omnidirectional Printing of PEDOT:PSS for High-Conductivity Spanning Structures. *ACS Appl. Mater. Interfaces*. **15**, 57717-57725 (2023)
5. Zhou, T. et al. 3D printable high-performance conducting polymer hydrogel for all-hydrogel bioelectronic interfaces. *Nat. Mater.* **22**, 895–902 (2023).
6. Yu, J. et al. 3D Printing of Robust High-Performance Conducting Polymer Hydrogel-Based Electrical Bioadhesive Interface for Soft Bioelectronics. *Small*, 2308778.
7. Hill, I. M. et al. Imparting High Conductivity to 3D Printed PEDOT:PSS. *ACS Appl. Polym. Mater.* **5**, 3989–3998 (2023).
8. Wang, F. et al. 3D Printed Implantable Hydrogel Bioelectronics for Electrophysiological Monitoring and Electrical Modulation. *Adv. Funct. Mater.* 2314471 (2023).
9. Lu, B. et al. Pure PEDOT:PSS hydrogels. *Nat. Commun.* **10**, 1043 (2019).
10. Chong, J. et al. Highly conductive tissue-like hydrogel interface through template-directed assembly. *Nat. Commun* **14**, 2206 (2023).

11. Won, D. *et al.* Digital selective transformation and patterning of highly conductive hydrogel bioelectronics by laser-induced phase separation. *Sci. Adv.* **8**, eabo3209.
12. Wei, H. *et al.* Orthogonal photochemistry-assisted printing of 3D tough and stretchable conductive hydrogels. *Nat Commun* **12**, 2082 (2021).
13. Zhang, J. *et al.* Engineering Electrodes with Robust Conducting Hydrogel Coating for Neural Recording and Modulation. *Adv. Mater.* **35**, 2209324 (2023).
14. Yao, B. *et al.* Ultrastrong, highly conductive and capacitive hydrogel electrode for electron-ion transduction. *Matter* **5**, 4407–4424 (2022).

Comment 8: How to calculate the “3D aspect ratio” should be clearly illustrated, such as which structure as well as detailed data on the height and width.

We sincerely appreciate the reviewer's feedback. A detailed method to calculate the “3D aspect ratio” is added in the revised supplementary information with a schematic illustration (Fig. R10).

Fig. R10 (Supplementary Fig. 27 in the revised manuscript) | Calculation method of 3D Aspect ratio

(Revised Main Text, Page 16, Line 9)

Fabrication of PILC printed 3D implantable devices

A poly(dimethylsiloxane) (PDMS; Sylgard 184, Dow Corning) prepolymer solution was printed and utilized as an insulating substrate under *in vivo* conditions. After printing the PDMS prepolymer solution, it was cured at 80 °C for 30 min in an oven (OV3-30, JEIO TECH). First, the PILC ink was directly 2D printed with high-resolution (50 μm) on the as-cured PDMS substrate to fabricate 3D implantable devices. Second, 3D PILC microneedle with high aspect ratio was directly printed onto 2D recording regions with customized g-code. Finally, PDMS encapsulation was coated on the 3D implantable devices except for recording regions. The printed aspect ratio was calculated as seen in **Supplementary Fig. 27**.

Comment 9: Since applications in the last two figures are all printed mostly in a 2D manner by using a 3D printer, without demonstrating the new capability or functionality of well-printed 3D structures. More applications should be demonstrated to harness the advantages of 3D structures.

We sincerely appreciate the reviewer's feedback and the valuable insights that can improve the quality of this work. As per reviewer's request, we have added new 3D bioelectronic applications (see response to comment 1) to our manuscript. In addition, the impedance and charge storage capacity were measured with 3D printed PILC electrodes. The 3D electrodes exhibited improved impedance (151Ω at 1kHz) and charge storage capacity (104.827 mC/cm^2) compared to 2D PILC electrodes ($>1 \text{ k}\Omega$ at 1kHz and 47.04 mC/cm^2) as 3D electrodes have increased electrochemical surface area (**Fig. R11**).

Fig. R11 (Supplementary Fig. 26 in the revised manuscript) | Impedance and Charge storage capacity (CSC) for 3D-printed and 2D printed PILC electrode. a, A plot to compare Impedance between 3D-printed and 2D-printed PILC electrodes. **b,** A plot to compare CSC between 3D-printed and 2D-printed PILC electrodes. The 3D printed PILC electrode demonstrates increased impedance (151Ω at 1kHz) and CSC (104.827 mC/cm^2) in comparison to 2D printed PILC electrodes ($>1 \text{ k}\Omega$ at 1kHz and 47.04 mC/cm^2).

(Revised Main Text, Page 11, Line 20)

Moreover, the printed PILC devices enabled low-voltage stimulation ($\sim 60\text{mV}$) of the sciatic nerve, demonstrating its use for safe electrical stimulation in vivo (**Fig. 6g, 6h, Supplementary Fig. 26**).

Comment 10: Some figures are unclear and may confuse the readership. The schematic drawing in Fig. 1b and Fig.1c needs a clear illustration between different particles and specific ink compositions. The Y-axis scale label of Fig. 2d may be better to be consistent with that of Fig. 2c. The abbreviation “3DP” in Fig. 5c and Fig. 6d requires clarification in the manuscript.

We sincerely appreciate the reviewer's feedback and the valuable insights that can improve the quality of this work. Fig. 1b and 1c are modified with the addition of clear illustrations between PILC colloidal particles and specific ink composites (Fig. R12). In addition, a new schematic illustration is added to the revised manuscript to emphasizing the versatile application of the PILC ink.

Moreover, the Y-axis scale label of Fig. 2d was modified to be consistent with that of Fig. 2c (Fig. R13). In addition, the abbreviation of “3DP” in Fig. 5c, that was relocated to Supplementary Fig. 19c, and Fig. 6d was clarified to 3D printed (Fig. R9, R14).

Fig. R12 (Fig. 1b-c in the revised manuscript) | PEDOT:PSS ionic liquid colloidal (PILC) ink for 3D printed bioelectronics b, Schematic illustrations of the PILC ink with minimal ionic liquid content and good structural integrity. c, Schematic illustrations of conventional PEDOT:PSS ionic liquid composites with cytotoxic ionic liquid components and poor structural integrity.

Fig. R13 (Fig. 2b-c in the revised manuscript) | Ionic liquid facilitated hydrogen-bonding of PILC ink for high resolution and high aspect ratio 3D printing. b, FT-IR spectra of the

PILC ink as hydrogen bonding progresses (measured while drying the PILC ink). **c**, Viscosity of PILC ink in comparison to conventional PEDOT:PSS ionic liquid composites.

Fig. R14 (Fig. 6d in the revised manuscript) | 3D printed PILC devices for implantable bioelectronics. **d**, Schematic (middle) and images of PILC implantable devices for optogenetic ECoG recording (left) and sciatic nerve electrical stimulation (right)

Reviewer #3

Overall comment: *This paper reports a new fabrication method for 3D printable, biocompatible, and highly conductive PEDOT:PSS-ionic liquid colloidal (PILC) inks that can be used in various bioelectronics applications, including the physiological detection (ECG and EMG) and in vivo optogenetic ECoG signal. Although this work shows significant outcomes, a few points should be clarified before publication.*

Specific comments: The author provides three major applications: omnidirectional printing, skin electrode printing, and in-vivo electrode printing. The mechanical study of the omnidirectional printed structure looks important to prove the ink's potential usage for 3D applications. However, the detailed experiments needed to be included. For skin-electrode and in-vivo electrode applications, they look more like 2D electrode applications, which have a lot of comparable examples reported previously. Please clarify how this work is different from others.

We thank the reviewer for the appraisal of our work and appreciate his/her valuable time to give helpful comments and suggestions. First, we conducted detailed experiments on the mechanical properties of the PILC ink (please refer to the response to comment 3 and **Table R1** and **R2** in the revised manuscript).

Ref	Viscosity (Pa*s)	Yield stress (Pa)	Storage modulus (Pa)	Conductivity (S/cm)	Intrinsic biocompatibility	3D-Printed Layers (aspect ratio)	Water content (%)	Printing resolution (um)
[1]	5*10 ²	10 ²	5*10 ³	28 (dry) ----- 155 (wet)	X (>24 hr post annealing)	20 (N/R)	87	30
[2]	23	N/R	N/R	0.022	N/A (> 1.5 hr post-treatment)	N/A	260	94
[3]	10 ²	10 ⁴	5*10 ³	858.1 (Post-acid treatment)	N/A	25 (N/R)	N/R	50
[4]	3*10 ³	4*10 ⁴	4*10 ³	72 (wet) ----- 654 (Post-acid treatment)	N/A	N/R (8~15)	N/R	15
[5]	100	N/R	N/R	11	N/A (> 24 hr post-treatment)	N/A	80	100
[6]	15	N/R	10 ³	0.0159	N/A (> 1 hr post-treatment)	N/A	N/R	150
[7]	10 ³	10 ³	330	1,200 (Post-acid treatment)	N/A	10 (N/R)	N/R	50
[8]	10 ²	10 ³	10 ³	0.09	N/A (> 24 hr post-treatment)	N/A	75	80
This Work	10 ⁶	10 ³	10 ⁵	286	O	>100 (>4)	28.75	50

N/A : Not Applicable
N/R : Not Reported

Table R1. (Supplementary Table. 1 in the revised manuscript) | Comparison with previous papers using printable PEDOT:PSS inks

Ref	Method	Young's modulus (kPa)	Tensile strength (MPa)	Stretchability (%)	Charge Storage Capacity (mC cm ⁻²)	Initial Conductivity (S/cm)
[1]	Nozzle printing	1,100	N/R	N/R	N/R	28 (dry) 155 (wet)
[2]	Liquid-in-Liquid Printing	41.2 - 3,875	1 - 2.5	76 - 800	N/R	0.022
[5]	Nozzle printing	1,000	0.5 - 3.5	400	6	11
[6]	Nozzle printing	5 - 65	110	349	12.37	0.0159
[8]	Nozzle printing	650	0.9	120	5.83	0.09
[9]	Photolithography	2,000	0.5 - 2.2	15 - 35	60	20-40
[10]	Molding	25	130	610	80	247
[11]	Laser-Induced phase separation	120	15	15	32.13	670
This Work	Nozzle printing	750	0.139	88.6	2D : 47.04 3D : 104.827	286

N/R : Not Reported

Table R2. (Supplementary Table. 2 in the revised manuscript) | Comparison with previous papers using PEDOT:PSS inks

(Revised Main Text, Page 4, Line 17)

The combination of unique material properties (high conductivity, high printing resolution, biocompatibility before post processing, and high 3D aspect ratio) exhibits the broad applicability of the PILC ink as a general purpose PEDOT:PSS ink in bioelectronic applications (**Fig. 1e-f, Supplementary Fig. 3 and Supplementary Table. 1-Table 3**).

The main contribution of this work is the versatile application of the PILC ink, ranging from 2D patterns to freestanding 3D structures. As the reviewer mentioned, many previous reports have demonstrated 2D electrode applications. To highlight the unique capability of our ink, PILC-based 3D bioelectronics were added as main figures in the revised manuscript, while moving the 2D electrode applications to the supplementary information (**Fig.5 in the previous version of the manuscript, Fig R1**).

The PILC 3D bioelectronics was used for neural interfaces in-vivo (**Fig R2**) where the device was capable of 1) recording from deep brain regions and 2) increase fidelity of evoked signals during recording. We also made other complex 3D structures with different aspect ratios (**Fig.**

R3). In addition, we also demonstrated the ability of the PILC ink to create a suspended PILC artificial 3D tissue structure (diagonal overhang: ~ 5 mm with various printing angle) (**Fig. R3**). This potentially enables 3D organoid-on-a-chip for organ-specific genetic disease monitoring system, where the PILC can bridge two different interfaces with a high vertical and horizontal gap.

Fig. R1 (Supplementary Fig. 19 in the revised manuscript) | 3D printed PILC electrodes for on-skin bioelectronics. **a**, Images of multi-electrode EMG array printed with the PILC ink (left) and subsequent transfer to wet skin (right). **b**, Impedance spectra of PILC EMG electrodes in comparison to commercial 3M electrodes. **c**, Schematic (left) and average ECG waveform of the PILC ECG electrodes (middle) and conventional 3M electrodes (right). **d**, Schematic (left) and EMG waveform (right) collected with PILC EMG electrodes during bicep contraction.

(Revised Main Text, Page 10, Line 5)

We demonstrate the use of PILC ink to fabricate a large area on-skin e-tattoo (**Supplementary Fig. 19a**). After printing the PILC ink on a super hydrophobic (polypropylene) flexible substrate, the PILC electrodes can be transferred on to wet skin under gentle pressure (**Supplementary Fig. 19a, Supplementary Fig. 20**) with adhesion strength of 1.884 N/mm between wet porcine skins and printed PILC films, which is need to peel off wet porcine skins attached with film of printed PILC (**Supplementary Fig. 21**). The high conductivity and conformal contact of PILC electrodes results in lower interfacial skin impedance (200 kΩ·mm² at 1 kHz) compared to that of the commercial 3M electrodes (8,000 kΩ·mm² at 1 kHz) (**Supplementary Fig. 19b**).

Utilizing the lower interfacial skin impedance, PILC electrodes demonstrate high quality recordings of both ECG (electrocardiogram) and EMG (electromyogram) waveforms. PILC electrodes demonstrated an increased signal amplitude in the ECG waveform in comparison with commercial 3M electrodes (**Supplementary Fig. 19c**). In addition, PILC

electrodes demonstrate high signal to noise ratios (16.80 dB) during EMG recordings of bicep muscles (**Supplementary Fig. 19d**).

Fig. R2 (Fig. 7 in the revised manuscript) | 3D PILC devices for flexible in vivo bioelectronics **a**, Schematic of 3D PILC arrays for fabrication of flexible, biocompatible 3D devices. 3D PILC arrays enable customizable interfacing with neural signals from deeper cortical layers and hippocampus, which are difficult to achieve with 2D surface arrays. **b**, Image of 3D PILC arrays enabled through high aspect ratio printing of PILC ink **c**, Power spectral density (PSD) of 3D and 2D PILC arrays for recording of characteristics endogenous neural signals from deeper brain regions (i.e., hippocampus). The red shaded area indicates the frequency band of theta oscillations (4~12 Hz). The theta band peaks and their harmonic frequencies in the PSD are marked with red arrows. **d**, Spectrogram of 3D (top) and 2D(bottom) PILC arrays. **e**, Raw local field potentials recorded from 3D PILC arrays (top) and 2D PILC arrays (bottom). **f**, Schematic of sensory stimulation of the secondary visual cortex in an in vivo mouse model. **g-i**, Evoked neural signals from a (g) 2D array during contralateral stimulation, (h) 3D array during contralateral stimulation, and (i) 3D array during ipsilateral stimulation.

(Revised Main Text, Page 12, Line 1)

We also demonstrated that the versatile printing of the PILC ink enables the fabrication of 3D bioelectronic devices, such as vertical spike arrays (**Fig. 7a-b**). These flexible and biocompatible devices consisting of array of PILC electrodes with high aspect ratio enables the recording of neural signals from deep brain regions, such as the hippocampus, which is difficult to achieve with 2D surface arrays. We validate that the recorded signals demonstrate characteristics of hippocampal signals by examining the signal frequency spectrum. A dominant signal characteristic of the hippocampus are theta wave oscillations, which are signals prominent in the 4~8Hz frequency band³⁶. The 3D PILC arrays demonstrates peaks at ~5 Hz and its harmonic frequencies as shown in the power spectral density (**Fig. 7c**), spectrogram (**Fig. 7d**), and raw signal waveforms (**Fig. 7e**). While in contrast, this peak is not observed in the 2D PILC arrays, due to their spatial restriction to the surface of the brain (**Fig. 7c-e**).

In addition, the 3D PILC arrays demonstrate high fidelity recording during electrophysiology of sensory evoked potentials (**Fig. 7f**). Cortical neural signals demonstrate maximal amplitude when neural probes are inserted 300~500 μ m below the cortical surface (i.e., Layer IV~V)³⁷. We target the secondary visual cortex, which has been shown to have reduced signal amplitude in comparison to the primary visual cortex upon application of visual stimuli.

Upon application of visual stimulus to the eye contralateral to the PILC array, signals-correlated to stimulus intensity were observed in the 3D PILC array in contrast to the 2D array (**Fig. 7g-h**). The correlation in visually-evoked signal amplitude and stimulus intensity verifies successful recording of sensory evoked potentials. In addition, to verify that the signal is not a recording artifact, ipsilateral visual stimulation is also conducted. No signal is observed upon ipsilateral stimulation, suggesting there is minimal light-induced artifacts (**Fig. 7i**).

References

36. Nuñez, Angel, and Washington Buño. The theta rhythm of the hippocampus: from neuronal and circuit mechanisms to behavior. *Front. cell. neurosci.* **15**, 649262 (2021).
37. Narcisse, Darryl, et al. Monitoring visual cortical activities during progressive retinal degeneration using functional bioluminescence imaging. *Front. neurol.* **15**, 750684 (2021).

(Revised Main Text, Page 19, Line 11)

In vivo recording of 2D and 3D endogenous electrocorticography signals and visually-evoked potentials

Surgery was conducted in the same manner as the optically evoked electrocorticography signals. A 2D or 3D PILC device was placed on top of the dura. During signal recording, the OEP signals are analog filtered with a bandpass filter of 3~300 Hz with a multichannel electrophysiology recording equipment (Lab Rat Ephys, Tucker-Davis Technologies).

For recording of visually-evoked potentials, a 2D or 3D PILC device was placed on top of secondary visual cortex (-1 mm ML; -3 mm AP). A blue laser (465nm IOS-465 Intelligent Optogenetics System, RWD) was utilized to given direct optical stimulation from a ~5cm distance to either the contralateral or ipsilateral eye. Waveforms were aligned at each

stimulation pulse onset and averaged to produce average waveforms during visual stimulation using MATLAB.

Fig. R3 (Fig. 5 in the revised manuscript) | 3D high aspect ratio printing of PILC inks with air printing w/o supported layers. a, Images of 3D high-aspect ratio printing of PILC inks. **b,** Images of air printing of PILC inks without supported layers. **c,** Images of 3D printed PILC structure that mimics pyramid structures. **d,** Images of an LED mounted on the 3D printed PILC structure.

(Revised Main Text, Page 9, Line 12)

The improved yield stress and storage modulus of the PILC ink, compared to pristine PEDOT:PSS inks and other PEDOT:PSS composites, enable the 3D high aspect ratio printing of PILC inks without any collapse of printed structures during printing (**Fig. 5a**). Additionally, due to its high yield stress and storage modulus, the PILC ink can be printed in air without any support layers, enabling the creation of bridges with the potential for 3D interconnections of individual circuits (**Fig. 5b**). Leveraging these capabilities for 3D high aspect ratio and air printability, an LED chip was mounted on a pyramid structure created using PILC inks after printing (**Fig. 5c**) and subsequently operated (**Fig. 5d**).

(Revised Main Text, Page 15, Line 16)

Fabrication of a 3D high aspect ratio circuit for operation an LED chip

To demonstrate the fabrication of a 3D high aspect ratio circuit, a SMD LED chip (2.6 V; HSMH-H170 with a 7 mm of printed height) was mounted on a pyramid PILC structure. The PILC ink was connected onto both the cathode and anode with silver paste and connected to

Teflon-wrapping wires (SME). To turn on the LED, a voltage of 2.6 V was applied to the LED chip through a source meter (Keithley 2400, Tektronix Inc.).

Fig. R4 (Supplementary Fig. 14 in the revised manuscript) | 3D diagonally printed circuits lines for achieving an artificial 3D tissue with artificial kidney made of Ecoflex-0020

(Revised Main Text, Page 9, Line 1)

The PILC ink can be utilized to print suspended interconnects by simultaneously moving the x- and z- stage in the air (**Fig. 4b-c**). In addition, 3D diagonally suspended circuit lines can be printed onto an artificial kidney (made of Ecoflex-0020, with a height of 7 mm), demonstrating the potential of employing a 3D interconnection of PILC inks via omnidirectional printing. This approach can effectively bridge the two different interfaces, even with a high angle and high gap, (as shown in **Supplementary Fig. 14**).

(Revised Main Text, Page 15, Line 5)

2D & 3D printing of PILC inks

A direct-ink-writing printer (BIO X6, CELLINK) was used to print the PILC inks onto various films. The as-prepared PILC ink was loaded into a 3mL syringe and was utilized for the pressure-driven extrusion of PILC inks during printing. The tip diameter of the nozzles and printing speeds were determined in accordance with the specified requirements (nozzle diameter: 30 μm to 250 μm , printing speed: 6.4 mm/s to 12 mm/s). For the 3D printing of PILC inks, such as omnidirectional printing, MATLAB software (R2019a, The MathWorks, Inc.) was utilized to customize the g-code and incorporated automatic x-stage movements (printing speed: 1.4 mm/s to 2.0mm/s with 30 Gauge nozzles), z-stage movements (printing speed: 1.4 mm/s to 2.0mm/s) and pressure control (50 - 100 kPa).

1. *How long does the ink drying time require for multilayer printing in Fig. 2h?*

To clearly define the ink drying time for multilayer printing, we measured the weight loss after printing. After 1 minute at 60°C, the PILC multilayer (>100) was fully dried with no changes in weight.

(Revised Main Text, Page 7, Line 12)

In addition, PILC ink can be utilized to print tall, customized three-dimensional structures. The PILC can be highly stacked with an aspect ratio of 4, which is relatively high compared to previous PEDOT:PSS-based inks which can be fully dried within 1 min at 60 °C (**Fig. 2f-h**).

2. The skin contact impedance at a low-frequency level (~1 Hz) should be monitored for physiological signal monitoring.

We thank the reviewer for this helpful suggestion. The skin contact impedance was measured with inclusion of a low frequency level (~ 1 Hz) at previous impedance spectra (**Fig. R5**).

Fig. R5 (Supplementary Fig. 19b in the revised manuscript). | Impedance spectra of PILC EMG electrodes in comparison to commercial 3M electrodes.

(Revised Main Text, Page 10, Line 11)

The high conductivity and conformal contact of PILC electrodes results in lower interfacial skin impedance ($200 k\Omega \cdot mm^2$ at 1 kHz) compared to that of the commercial 3M electrodes ($8,000 k\Omega \cdot mm^2$ at 1 kHz) (**Supplementary Fig. 19b**).

3. What is the adhesion strength of skin electrodes? In Supplementary Fig. 12, what happened to the electrode with additional strain? Maximum stretchable range? Repeatability?

We thank the reviewer for bringing this issue to improve the quality of this work. Adhesion strength was measured by attaching printed PILC film between two wet porcine skins. The results indicate that an adhesive strength of 1.884 N/mm is required to peel off the two porcine skins attached with the printed PILC film (Fig. R6). Furthermore, in response to your inquiry, additional experiments were conducted to examine the mechanical properties of PILC electrodes. In contrast to freestanding PILC film, which can be stretched up to 10% strain, printed PILC electrodes on a soft substrate (PDMS) is stretchable up to 90% strain and has cyclability up to 70% strain for 1,000 cycles (Fig. R7-R9). Such an effect can be attributed to strain energy dissipation as previously reported^{1,2}

Reference

[1]: Yang, J. C. et al. Geometrically engineered rigid island array for stretchable electronics capable of withstanding various deformation modes. *Sci. Adv.*, 8, eabn3863 (2022).

[2]: Cai, M., Nie, S., Du, Y., Wang, C. & Song, J. Soft Elastomers with Programmable Stiffness as Strain-Isolating Substrates for Stretchable Electronics. *ACS Appl. Mater. Interfaces*, 11, 14340–14346 (2019).

Fig. R6 (Supplementary Fig. 21 in the revised manuscript) | Adhesion property of PILC between wet porcine skins.

(Revised Main Text, Page 10, Line 6)

After printing the PILC ink on a super hydrophobic (polypropylene) flexible substrate, the PILC electrodes can be transferred on to wet skin under gentle pressure (Supplementary Fig. 19a,

Supplementary Fig. 20) with adhesion strength of 1.884 N/mm between wet porcine skins and printed PILC films, which is need to peel off wet porcine skins attached with film of printed PILC (**Supplementary Fig. 21**).

Fig. R7 (Supplementary Fig. 16 in the revised manuscript) | Mechanical properties of freestanding PILC films. **a**, A photograph of the set-up used to measure the mechanical properties of the PILC ink on water. **b**, Stress versus strain of the PILC film. Printed PILC films demonstrate low Young's modulus (750 kPa) similar to tissues.

(Revised Main Text, Page 9, Line 22)

The unique properties of the PILC ink also makes it an ideal material for rapid on-demand fabrication of on-skin bioelectronics for health monitoring applications. As electrodes printed from PILC ink have similar mechanical properties (Young's modulus 750 kPa, **Tensile Strength 139.3 kPa, and Toughness 4.0 kJ/m³** as seen in **Supplementary Fig. 16**) with biological tissues,³³ PILC electrodes can be utilized as soft electrodes to interface with tissue.

Fig. R8 (Supplementary Fig. 17 in the revised manuscript | Mechanical properties of printed PILC on a PDMS substrate at strain rate: 50 mm/min.

Fig. R9 (Supplementary Fig. 18 in the revised manuscript) | Repeatability of printed PILC on a PDMS substrate for 1,000 cycles @ 70% strain at strain rate: 50 mm/min).

(Revised Main Text, Page 10, Line 2)

Furthermore, printed PILC electrodes on soft substrate (PDMS) have stretchability up to 90% strain and cyclability at 70 % strain for 1,000 cycles (**Supplementary Fig. 17 and Fig. 18**) in contrast to freestanding PILC film, which can only be stretched up to 10% strain. This can be attributed to energy dissipation mechanism as previously reported^{34,35}.

References

34. Yang, J. C. et al. Geometrically engineered rigid island array for stretchable electronics capable of withstanding various deformation modes. *Science Advances* 8, eabn3863 (2022).
35. Cai, M., Nie, S., Du, Y., Wang, C. & Song, J. Soft Elastomers with Programmable Stiffness as Strain-Isolating Substrates for Stretchable Electronics. *ACS Appl. Mater. Interfaces* 11, 14340–14346 (2019).

4. In Fig. 6a, what is the bending radius?

We sincerely appreciate this comment. The bending radius was added as Fig. 6a (R: 1.5 mm) (Fig. R10).

Fig. R10 (Fig. 6a in the revised manuscript) | 3D printed PILC devices for implantable bioelectronics. a, Change in resistance versus bending cycles of flexible PILC devices (Bending radius is 1.5 mm).

5. There needs to be more description of why the performance in Fig. 6a-c is important for the in-vivo electrode application.

Thank you for valuable comment. We have added additional explanations on how Fig. 6a-c is important for the in-vivo electrode application as shown below.

(Revised Main Text, Page 10, Line 21)

“In addition to on-skin bioelectronics, the PILC ink can be utilized to fabricate PILC devices for implantable bioelectronics. In order to for PILC devices to be utilized in in vivo environments, the PILC electrodes should be able to handle mechanical and electrical stress during surgical insertion and potential animal motor responses (i.e., leg movement). To verify the mechanical stability, a cyclic bending test is conducted with a bending radius of 1.5 mm (Fig.6a). The PILC electrode demonstrate minimal change in resistance during 10,000 bending cycles, demonstrating its tolerance to mechanical deformation during in vivo electrode application. Furthermore, the PILC electrical properties, demonstrates high conductivity in saline and tolerance to wet physiological conditions, enabling its application for electrophysiological recording (Fig.6b, Supplementary Fig. 22). In addition, the high charge storage capacity of the PILC electrode, in comparison to conventional metals such as gold, demonstrate its efficient charge injection in physiological environment and opens potential for utilizing lower stimulation voltages, and thereby safer stimulation conditions in vivo (Fig.6c, Supplementary Fig. 23).”

6. In Fig. 6e-f, the author said PILC ECoG device can record ECoG signals with high SNR and spatial selectivity. How do you validate this performance?

Thank you for your comment on the ECoG device performance. To validate the high SNR, we calculated the SNR and RMS noise at the optical stimulation electrode. An SNR of 9.0 was measured during optical stimulation and an RMS noise signal of $\sim 16\mu\text{V}$ was recorded prior to stimulation.

To validate the spatial selectivity, we compared the amplitude of the optically-evoked potential at the electrode placed directly below the optical fiber versus an adjacent electrode. There is a ~ 5 fold increase in amplitude at the electrode directly on the stimulation site compared to an electrode within 1.1 mm of the stimulation site. While there is a slight signal as the adjacent electrode due to the spreading of the light from the tip of the optical fiber, the signal amplitude is significantly decreased ($-56\mu\text{V}$ 1.1mm from the stimulation site versus $-290\mu\text{V}$ at the stimulation site) as shown in the Figure R11.

The additional data on SNR and spatial selectivity are added to the revised manuscript as follows:

(Revised Main Text, Page 11, Line 16)

“A multi-channel PILC ECoG device enabled the recording of optically-evoked ECoG signals in a Thy1-Chr2 transgenic mouse with high SNR (SNR 9.0) (Fig.6e, 6f). Furthermore, the PILC ECoG device demonstrates high spatial-selectivity with a high signal amplitude at the site of optical stimulation ($-290\mu\text{V}$ at an electrode 0 mm from optical stimulation) compared an adjacent electrode ($-56\mu\text{V}$ at an electrode 1.1mm from optical stimulation) (Supplementary Fig. 25).”

Fig. R11 (Supplementary Fig. 25 in the revised manuscript) | Spatial selectivity of ECoG optically evoked potentials in Thy1-Chr2 transgenic mice.

7. If applicable, please provide photo images of the electrode attached to the mouse's brain or nerves so that it can improve the manuscript quality.

We thank the reviewer for their kind suggestion. In order to improve the quality of the manuscript, we have added a photo of the PILC electrodes attached to the mouse nerve (Fig. R12).

Fig. R12 (Supplementary Fig. 24 in the revised manuscript) | Image of the PILC electrode utilize to interface with the sciatic nerve in an in vivo mouse model.

We thank the reviewer for their insightful comments and the time spent for their thorough review of our work. We have worked to fully address these comments with additional data that have substantially improved the manuscript.

[Additional Modifications]

Modification 1:

We have changed a title of this work from “3D Printable and Biocompatible PEDOT:PSS-Ionic Liquid Colloids with High Conductivity for Rapid On-demand Fabrication of Bioelectronics” to “3D Printable and Biocompatible PEDOT:PSS-Ionic Liquid Colloids with High Conductivity for Rapid On-demand Fabrication of 3D Bioelectronics”.

Modification 2:

We have modified abstract section to revised manuscript.

[Revised Main Text, Page 2]

3D printing has been widely used for on-demand prototyping of complex three-dimensional structures. In biomedical applications, PEDOT:PSS has emerged as a promising material in versatile bioelectronics due to its tissue-like mechanical properties and suitable electrical properties. However, previously developed PEDOT:PSS inks have not been able to fully utilize the advantages of commercial 3D printing due to its long post treatment times, difficulty in high aspect ratio printing, and low conductivity. We propose a one-shot strategy for the fabrication of PEDOT:PSS ink that is able to simultaneously achieve on-demand biocompatibility (no post treatment), structural integrity during 3D printing for tall three-dimensional structures, and high conductivity for rapid-prototyping. By using ionic liquid-facilitated PEDOT:PSS colloidal stacking induced by centrifugal protocol, a viscoplastic PEDOT:PSS-ionic liquid colloidal (PILC) ink was developed. PILC inks exhibit high-aspect ratio vertical stacking, omnidirectional printability for generating suspended architectures, high conductivity (~286 S/cm), and high-resolution printing (~50 μm). We demonstrate the on-demand, versatile applicability of PILC inks through the fabrication of 3D circuit boards, on-skin physiological signal monitoring e-tattoos, and implantable bioelectronics (opto-electrocorticography recording, low voltage sciatic nerve stimulation and 3D vertical spike array for brain).

Modification 3:

We have added an Institutional Review Board (IRB) protocol number to revised manuscript.

[Revised Main Text, Page 18, Line 8]**Experiments on human subjects**

Ex-vivo experiments on e-skin electronics were performed under approval from the Institutional Review Board at Korea Advanced Institute of Science and Technology (protocol number: KH2023-153). All subjects voluntarily involved in experiments after informed consent.

Modification 4:

We have added a comment that informs about the sample size of the experiment.

[Revised Supplementary Information, Page 3]

Supplementary Fig. 2 | Optimized conductivity with various ionic liquid (IL) concentrations. The conductivity of the PILC ink increased with increasing IL concentration. **Data represents as mean values \pm SD (n = 4 independent experiments).**

[Revised Supplementary Information, Page 17]

Supplementary Fig. 13 | Biocompatibility of the PILC ink after centrifugal purification. Cell viability of the PILC ink after centrifuging one and two times. DMSO is plotted as a negative control group. **Data represents as mean values \pm SD (n = 3 independent experiments).**

[Revised Supplementary Information, Page 25]

Supplementary Fig. 23 | Charge storage capacity (CSC) for printed PILC electrodes in comparison to conventional metal electrodes. **a**, CSC of printed PILC electrodes over 1,000 cycles. **b**, A plot to compare CSC between printed PILC electrodes and gold electrodes. The PILC electrode demonstrates increased CSC (47.04 mC/cm²) in comparison to the gold electrode (0.0583mC/cm²). **Data represents as mean values \pm SD (n = 3 independent experiments).**

REVIEWERS' COMMENTS

Reviewer #2 (Remarks to the Author):

The authors have substantially revised the original manuscript and carefully clarified the novelty. All my concerns have been well addressed. I recommend acceptance of this work in NC.

Reviewer #3 (Remarks to the Author):

The revised manuscript shows an improved quality by reflecting the comments from reviewers. I suggest to accept this manuscript.

Subject: Revised Manuscript, “3D Printable and Biocompatible PEDOT:PSS-Ionic Liquid Colloids with High Conductivity for Rapid On-demand Fabrication of 3D Bioelectronics”

Reviewer’s comments are *italicized*, authors’ responses are in blue, and text quotes from the original and revised manuscript are in red.

REVIEWERS' COMMENTS

Reviewer #2 (Remarks to the Author):

The authors have substantially revised the original manuscript and carefully clarified the novelty. All my concerns have been well addressed. I recommend acceptance of this work in NC.

Reviewer #3 (Remarks to the Author):

The revised manuscript shows an improved quality by reflecting the comments from reviewers. I suggest to accept this manuscript.

We thank all the reviewers for their insightful comments and the time spent for their thorough review of our work.